# *In silico* EsxG EsxH rational epitope selection: Candidate epitopes for vaccine design against pulmonary tuberculosis

Constanza Estefania Martinez-Olivares[1], Rogelio Hernández-Pando[1,2]*, Edgar Mixcoha[3,4]*

**1** Experimental Pathology Department, Experimental Pathology Laboratory, Instituto Nacional de Ciencias Médicas y Nutrición "Salvador Zubirán", Mexico City, Mexico, **2** Facultad de Medicina, Pathology Department, Universidad Nacional Autónoma de México, Mexico City, Mexico, **3** Addiction Immunology Laboratory, Instituto Nacional de Psiquiatría "Ramón de la Fuente Muñiz"—CONACyT, Mexico City, Mexico, **4** Faculty of Sciences, Cellular Biology Department, Universidad Nacional Autónoma de México, Mexico City, Mexico

* emixcohahe@conacyt.mx (EM); rhdezpando@hotmail.com (RHP)

**Data Availability Statement:** https://figshare.com/projects/In_silico_EsxG_EsxH_Rational_Epitope_prediction/141302.

## Abstract

Rational design of new vaccines against pulmonary tuberculosis is imperative. Early secreted antigens (Esx) G and H are involved in metal uptake, drug resistance, and immune response evasion. These characteristics make it an ideal target for rational vaccine development. The aim of this study is to show the rational design of epitope-based peptide vaccines by using bioinformatics and structural vaccinology tools. A total of 4.15 μs of Molecular Dynamics simulations were carried out to describe the behavior in solution of heterodimer, single epitopes, and epitopes loaded into MHC-II complexes. In order to predict T and B cell epitopes for antigenic activation, bioinformatic tools were used. Hence, we propose three epitopes with the potential to design pulmonary tuberculosis vaccines. The possible use of the proposed epitopes includes subunit vaccines, as a booster in BCG vaccination to improve its immune response, as well as the generation of antibodies that interfere with the *Mycobacterium tuberculosis* homeostasis, affecting its survival.

## Introduction

Tuberculosis (TB) is the leading bacterial cause of death in the world. *Mycobacterium tuberculosis* (Mtb), the pathogen responsible for the infection, spreads by air route into droplets or aerosol sprays from sick people, affecting mainly the lungs (pulmonary TB). With the COVID-19 pandemic, the number of TB deaths increased by 5.6%, reversing the existing progress in reducing mortality [1]. *Mycobacterium bovis* Bacillus Calmette-Guérin (BCG) is the only available vaccine against TB. BCG vaccine was developed 100 years ago, it shows efficient protection against miliary and meningeal TB, but its protection against pulmonary TB ranges from 0 to 80% [2].

Vaccination is one of the most successful public health initiatives to prevent infectious diseases [3]. For TB disease, the development of new vaccines is necessary to reduce its worldwide

**Funding:** C.E.M.-O. is a doctoral student from the Programa de Maestría y Doctorado en Ciencias Bioquímicas, Universidad Nacional Autónoma de México (UNAM), and received a fellowship (413957) from Consejo Nacional de Ciencia y Tecnología (CONACyT) (CVU 406611). This work was also financially supported by Laboratorio Nacional de Computo de Alto Desempeño (LANCAD) project (21-2019 and 13-2020) awarded to EM. The funders had no role in study design, data collection, and analysis, decision to publish, or preparation of the manuscript.

**Competing interests:** The authors have declared that no competing interests exist.

incidence; hence, a rational design of vaccines is imperative. Rational vaccine design makes use of bioinformatics tools and computational chemistry approaches to identify protein regions based on physicochemical and structural information to trigger a protective immune response. The main benefit of these techniques is that they make it easier and less expensive to obtain proteins [3, 4] with improved and long-lasting immunological properties. [5]. With the use of these tools, the principle of the 3 Rs (replace, reduce and refine) is applied to the development of vaccine targets, and their prediction of efficiency and safety [6]. In recent years, the use of bioinformatics tools has allowed the design and selection of vaccine candidates for coronavirus [6, 7], dengue [8], *Mycoplasma pneumonia* [9], malaria [10], among many others.

One of the vaccine approaches is the use of epitope vaccines [11]. Epitope-based vaccines are considered safe because they exclude allergens, toxins, or functional domains of the pathogen. Therefore, the involved immune response is restricted to defined antigenic regions [3]. A disadvantage of subunit vaccines constituted by epitopes is the duration of the evoked immune memory due to their small size, short half-life, limited activity on immune cell stimulation, and a restricted immunological memory. Although BCG is an attenuated vaccine, it shows a limited activation of immune memory, but it is possible to extend it by using subunit vaccines as boosters after BCG vaccination. Vaccination is also necessary to stimulate crucial immune targets from innate immunity, as well as acquired or adaptive cellular and humoral immunity. BCG is efficient to stimulate innate immunity, being macrophage activation, and training immunity a clear example [12]. BCG stimulates adaptive cellular immunity, but it seems that it induces a limited activation of humoral immunity. Antibodies also contribute to the protection against Mtb, being essentially mediated by bacterial opsonization, and improving its intracellular killing [13]. B lymphocytes can produce bactericidal antibodies; it is quite possible that those antibodies neutralize or interfere with bacterial proteins related to essential metabolic processes, and in that way become real bactericidal antibodies.

The selection of an optimal antigen is a critical step in rational vaccine design [3]. Early secreted antigens (Esx) G and H from Mtb were selected for this work. Mainly, EsxG EsxH complex belongs to the ESX-3 secretion system, which is essential for bacteria viability, and it is metabolically important due to its role in iron uptake and in the adaptation to a low zinc environment (metal homeostasis) [14–17]. EsxG and EsxH proteins are secreted by the ESX-3 secretory system as a heterodimer. EsxG EsxH heterodimer not only has significant activity in bacterial metabolism, but it also has a remarkable immunological activity response to TB [18, 19]. Phagolysosome formation is a basic mechanism for bacterial elimination in phagocytic cells and is efficiently blocked by Mtb, after interacting with the host endosomal sorting complex (ESCRT) that prevents lysosome fusion. The ESCRT complex classifies the ubiquitin-labeled surface receptors to be degraded in the lysosome and loads peptides derived from the pathogens into MHC class II molecules. The EsxG EsxH complex inhibits phagosome maturation by binding Hrs protein and delaying T cell activation during infection. Therefore, it is important to identify mechanisms that promote a rapid and robust activation of protective T cells [20–23]. Furthermore, the genes that encode EsxG and EsxH proteins have been associated with Mtb first-line drug resistance mechanisms [24]. These characteristics make EsxG and EsxH proteins an ideal target for rational vaccine development.

The structure of the EsxG EsxH complex has been solved by Nuclear Magnetic Resonance. The heterodimer folds into a helix-turn-helix-turn structure of each monomer, which are arranged antiparallel to each other, forming a complex of four helices according to Protein Data Bank PDB ID: 2KG7 [25]. These helices are defined as follows. EsxG: residue 16 to 40 forms helix g1a, residues 41 to 42 forms helix g1b, and residues 48 to 77 form helix g2. EsxH: residue 18 to 40 forms helix h1a, residue 41 to 44 forms helix h1b, residues 61 to 72 forms helix h2a, and residues 75 to 81 forms helix h2b. The putative $Zn^{2+}$ binding site has been

identified in H14, H70, and H76 residues of EsxH protein, and the side chain of E77 is hypothesized to form the fourth $Zn^{2+}$ coordination group [25]. The EsxG EsxH complex model under code PDB ID: 2KG7 lacks $Zn^{2+}$; there are no other experimental coordinates of dimer that contain $Zn^{2+}$.

The aim of this study is to show the rational design of epitope-based peptide vaccines by using bioinformatics and structural vaccinology tools. Three epitopes of the EsxG EsxH complex were selected. A total of 4.15 μs of Molecular Dynamic simulations (MD) were used to describe the behavior of epitopes in solution or loaded into a MHC-II complex. B cell epitopes were predicted for humoral activation. Even knowing that Mtb is an intracellular pathogen, it must have entered from outside, so if the immune system has developed specific and avid antibodies against any of the TB proteins, it will be advantageous to inhibit the infection. Furthermore, if one of these proteins has demonstrated to be necessary for the survival of the bacteria, such as the EsxG EsxH complex, then the antibodies generated by active immunization with a rationally designed conjugate vaccine could protect against TB infection. Hence, we propose three promising candidate epitopes for vaccine design against pulmonary TB.

## Methods

### MHC class II binding epitope prediction

Major histocompatibility complex (MHC) class II binding epitope prediction was carried out by the ProPred (http://crdd.osdd.net/raghava/propred/) [26] and the NetMHCII 2.3 (https://services.healthtech.dtu.dk/service.php?NetMHCII-2.3) [27] servers. In order to locate binding regions with default parameters, 51 and 25 HLA-DR alleles were selected, respectively.

### B cell epitope prediction

A Linear B cell epitope prediction from EsxG (UniProt ID: O53692 [25]) and EsxH (UniProt ID: P9WNK3 [25]) sequences were carried out by BepiPrep 2.0 server (https://services.healthtech.dtu.dk/service.php?BepiPred-2.0) [28], using a threshold of 0.46. The DiscoTope 2.0 server was used for discontinuous B cell epitope prediction [29], with default parameters. The antigenic determinant prediction was performed by the IEDB analysis resource server (http://tools.iedb.org/bcell/) using the Kolaskar and Tongaonkar antigenicity scale [30], with default parameters.

### EsxG EsxH complex molecular dynamics simulation

A quaternary structure formed by tuberculosis virulence factors EsxG and EsxH (PDB ID: 2KG7) [25] was obtained from RCSB Protein Data Bank (http://www.rcsb.org) [31]. For this study, out of thirty conformers fetched from PDB, the first model was selected. Protein coordinates were centered in a dodecahedron box with a 1.5 nm distance between protein and unit cell walls. AMBER99SB was used as a force field to describe the whole system [32]. Protein was solvated using the SPC water model [33]. The system was neutralized by replacing water molecules until reaching 0.15 M NaCl mimicking physiological conditions. The neutralized system was energetically minimized for 1000 steps using the steepest descent algorithm. The system was equilibrated at 310 K with an NVT assembly for 1 ns. Subsequently, 5 ns of an NPT equilibration at 310 K and 1 atm of pressure was used. At the end of the equilibration stages, Molecular Dynamics simulations were carried out using the GROMACS software package version 5.1 [34, 35]. The Parrinello-Rahman barostat [36] and the Nosé-Hoover thermostat [37] was used to maintain constant pressure and temperature. Long-range electrostatic interactions were calculated using the particle-mesh Ewald algorithm [38], under periodic boundary

conditions, with a 0.15 nm grid. We simulated the system for 500 ns, motion equations were integrated every 2 ps, coordinates and velocities were saved every 10 ps.

To study the monomer's molecular behavior in solution, 250 ns MD simulations of each monomer were calculated at 310 K under the same conditions. Monomers coordinates were taken from 2KG7 model 1.

### Thermal unfolding simulations

A thermal unfolding pathway [39] was simulated to understand the relationship between the structure and thermal stability of the EsxG EsxH complex. The system was treated according to the above section. Simulations were conducted for 50 ns at 350, 400, 450, 500, and 550 K, respectively. The initial structure was the same for all simulated systems and corresponds to the structure after the final NVT simulation.

### Molecular docking

MHC class II molecules are crucial for initiating an antigenic-specific immune response to presented antigens [40], as well as for the activation, proliferation, and differentiation of B cells [41]. A peptide-protein docking of the selected epitopes was performed to predict their binding conformation into MHC class II molecules.

The crystal structure of the human leukocyte antigen (HLA) class II histocompatibility DRB1 allele (PDB ID: 6BIZ) [42] was obtained from RCSB Protein Data Bank [31]. HLA-DRB1 molecule was selected because it binds to most Mtb epitopes and is the allele with the highest superficial expression [43]. Predicted epitope sequences and HLA-DRB1 crystal structure were entered in the CABS-Dock server (http://biocomp.chem.uw.edu.pl/CABSdock) [44] for epitope-protein docking, so to obtain coordinates of the complexes. The most populated cluster model for each complex with the smallest RMSD value was selected. To improve the quality of interactions and peptide presentation, a flexible docking was performed in Flex-PepDock server (http://flexpepdock.furmanlab.cs.huji.ac.il) [45], using default parameters; the best resultant model was selected to perform MD simulations.

### MD simulations: Epitopes and HLA-epitopes complexes

Coordinates of the four chosen epitopes were selected and extracted from the 2KG7 file (model 1). MD simulations of epitopes were carried out along 600 ns for each epitope, under the same conditions described above. MD simulations of HLA-epitopes complexes were carried out along 250 ns under the same conditions. Motion equations were integrated every 2 ps, coordinates and velocities were stored every 10 ps.

### Molecular dynamics analysis

Analysis of unfolding trajectories was done using GROMACS package tools [34, 35]. The following parameters were analyzed: Root Mean Square Deviation (RMSD), Root Mean Square Fluctuation (RMSF), Radius of Gyration (RoG), Hydrogen bonds (Hb), secondary structure time evolution, Solvent Accessible Surface Area (SASA), clusterization, distance matrix, and Principal Component Analysis (PCA). Clusterization was performed using the gromos method [46], using RMSD cutoff 0.2 nm over the backbone atoms. Hydrogen bond analysis was calculated when donor and acceptor atoms were at a 0.3 nm distance, forming an angle of 180˚ ± 30˚ for at least 10 ps. Alignments for RMSD, RMSF, RoG were performed using backbone atoms, PCA was performed using alpha carbons coordinates of residues. Images were made in PyMOL [47] molecular visualization software and Gnuplot graph package [48].

The entire pipeline is shown graphically in Fig 1. Each step, server, and algorithm used to obtain T and B highly immunogenic epitopes are clearly explained.

## Results

### MHC class II and B cell epitope prediction

Considering that TB is an intracellular infection in which antigen presentation mediated by dendritic cells and macrophages is quite important to activate cell-mediated immunity, the

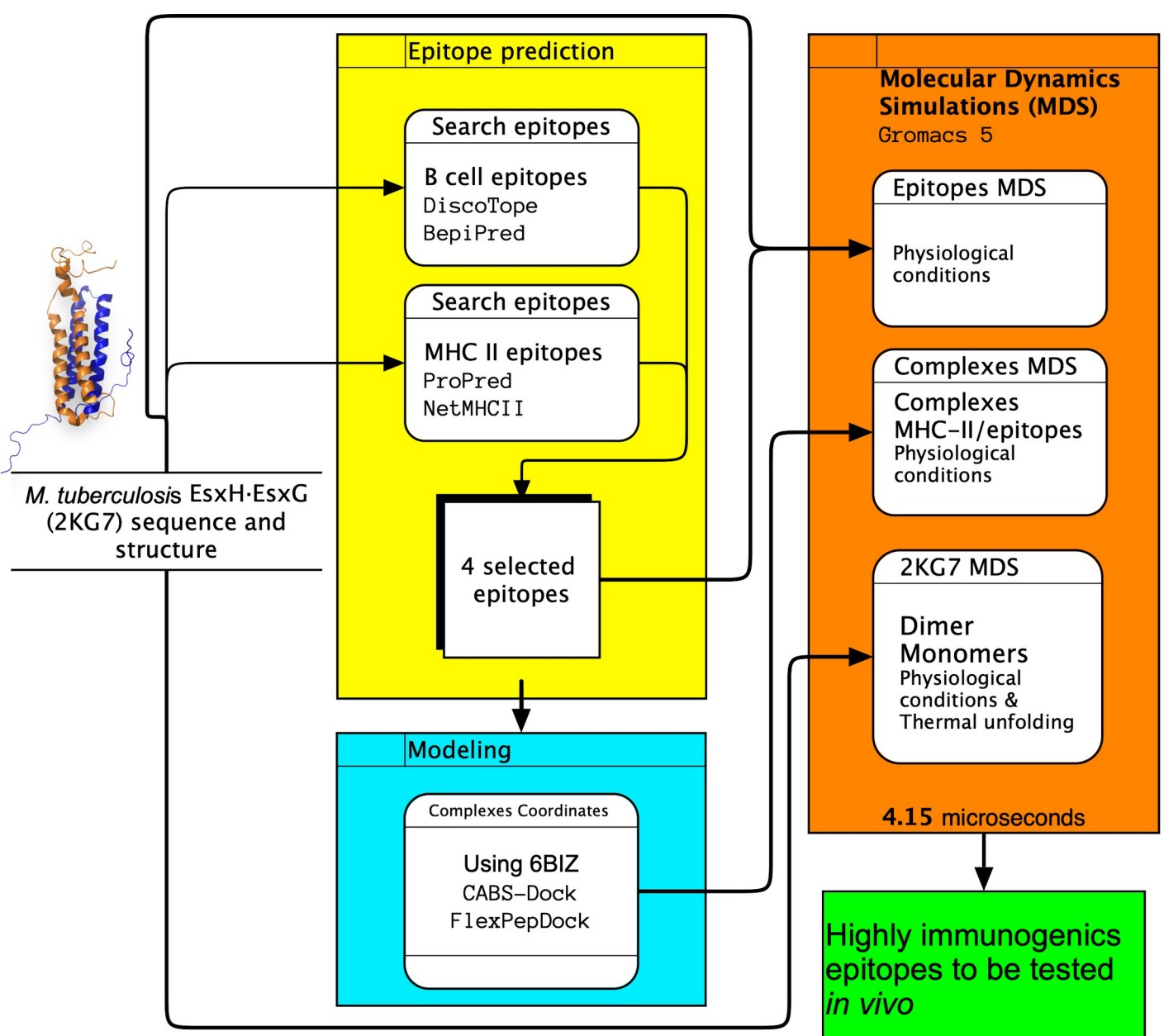

**Fig 1. Summary fig.** It shows all the steps that were applied in this work for the rational vaccine design to get highly immunogenic epitopes from EsxG and EsxH mycobacterial proteins. These are efficiently recognized by MHC-II and induce antibodies production using bioinformatics tools and Molecular Dynamics Simulations.

prediction of MHC class II epitopes is a crucial element. In this regard, ProPred [26] and NetMHCII 2.3 [27] threw seven epitopes from EsxG protein and fourteen epitopes for EsxH protein. Promiscuous regions that could bind to several HLA-DR alleles (S1 Table) were selected to identify residues that were bound to the MHC class II nucleus and nonamers epitopes.

One important aspect for an efficient anti-mycobacterial immune response is the production of antibodies that not only contribute to promoting macrophage opsonization, but they can mediate the neutralization of these proteins and induce substantial metabolic abnormalities during iron and zinc bacterial deprivation. To that respect, it has been demonstrated that, during active TB, there are high antibody titers against some specific mycobacterial compounds that correlate with the decrease of disease severity. This is the case of antibodies against the glycolipid lipoarabinomannan [49], or the Mtb phosphate transporter PstS1 [50]. Interestingly, the functional characterization of monoclonal antibodies anti-PstS1 isolated from TB patients directed against two different epitopes on PstS1 exhibit direct antibacterial activity [51]. Thus, the prediction of linear B cell epitopes was important. BepiPrep 2.0 prediction [28] resulted in four epitopes for EsxG, with more than five residues in length, and three residues located in positions 67, 68, and 79. For EsxH, three epitopes were predicted with more than sixteen residues in length. As shown in Fig 2A, the epitopes were homogeneously located along the sequence of both proteins.

Discontinuous B cell epitope prediction in DiscoTope 2.0 [29] yielded five residues for EsxG protein: [M[1], S[2], L[4], F[97]]; and the following residues for EsxH protein as components of discontinuous B-cell epitopes: [[1]MSQIMYNYPAM[11], H[14]], [[44]QGDTG[48], T[50]], [[79]NT[80],[83]MMARD[87]], [E[90], A[92], [94]WGG[96]] (Fig 2B).

Kolaskar and Tongaonkar's prediction, a semi-empirical method that uses physicochemical properties of amino acids was used to predict antigenic determinants [30]. Protein regions that satisfied the threshold value (antigenic propensity threshold 1.00) were predicted to be potential antigenic sites in which antibodies elicit antigen-specific responses (Fig 2C).

## Epitope selection

We selected three epitopes based on promiscuity to bind MHC class II molecules, antigenic propensity, structural stability, linear and discontinuous prediction of B cells, and surface exposition. According to this selection criteria, the epitope G1 ([11]LVASQSAFA[19]) was selected from EsxG protein. Epitopes H1 ([5]MYNYPAMLG[13]) and H2 ([65]LVRAYHAMSSTHE[77]) were chosen from EsxH protein. H2 epitope with thirteen residues joint three nonameric epitopes, included the zinc-binding region [25], Table 1. One epitope from EsxG protein was selected, which did not meet the selection criteria. Epitope G2 ([51]FQAAHARFVAAAAKVN[66]) was used to prove that our exclusion criteria were suitable for the selection.

## EsxG EsxH complex molecular dynamics simulation

To understand EsxG EsxH complex behavior in an aqueous solution, a MD simulation for 500 ns was performed. RMSD results showed that abrupt changes occurred from 0 to 9 ns, which might be related to the equilibration phase. The dimer presented two global conformations corresponding to two phases observed in the RMSD. The first one of them from 10 ns to 148 ns, the second one from 149 ns until 500 ns, keeping an average RMSD value of 0.995 nm along the simulation (S1A Fig). The temporary evolution of RoG was evaluated to understand the dimer's compactness degree. The system started with a RoG value of 2.245 nm, after 25 ns it fluctuated between 2.237 nm and 1.835 nm. At 149 ns the molecular compactness increased, it maintained an RMSD average of 1.887 nm (S1B Fig). These fluctuations coincided with the

## A. Linear B-cell epitope prediction

### EsxG

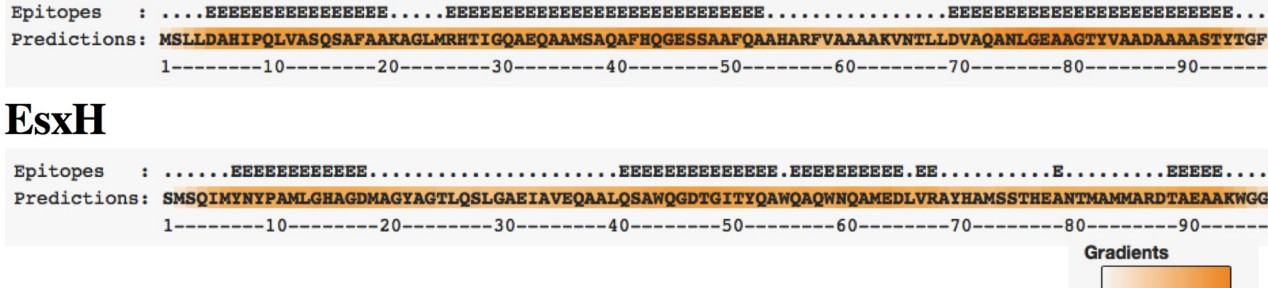

### EsxH

## B. Discontinuous B-cell epitope prediction

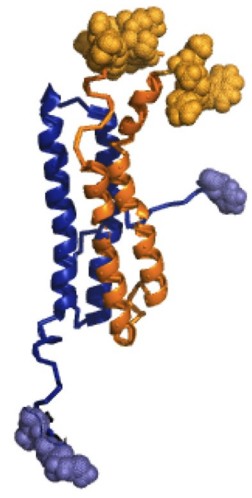

## C. Antigenic determinants prediction

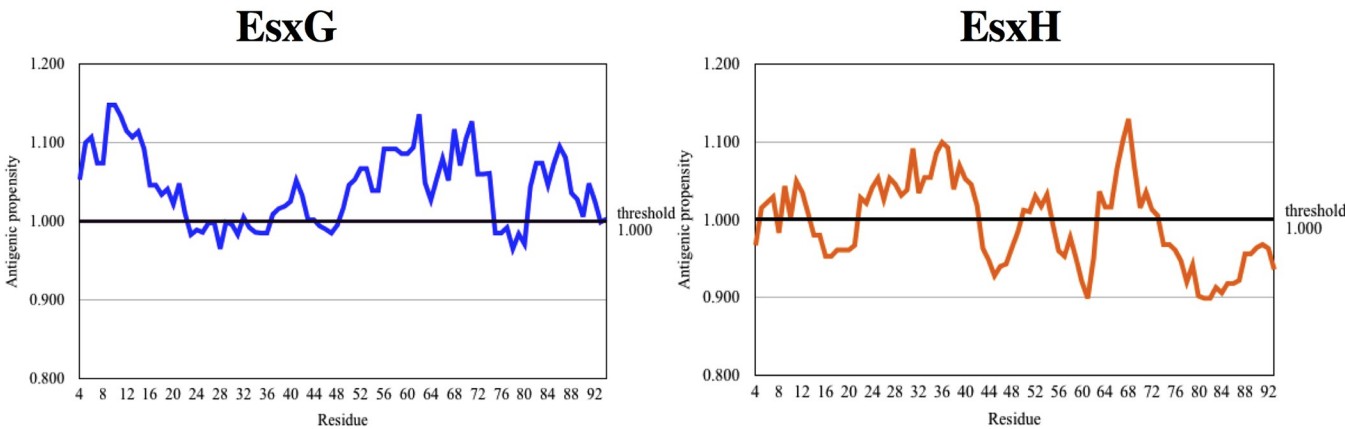

**Fig 2. Epitope prediction.** (A). Linear B cell epitope prediction. Prediction in BepiPred 2.0. Residues that present a score higher than 0.46 are indicated with an E at the top of the sequence. Residues with values greater than 0.46 are colored in dark orange. (B). Discontinuous B cell epitope prediction. Prediction in DiscoTope 2.0. Epitopes predicted are shown as spheres. (C). Antigenic determinant prediction. Kolaskar and Tongaonkar´s prediction method was employed; x-axis—residue number; y-axis—antigenic propensity. Residues with antigenic propensity greater than 1.000 were predicted as potential antigenic sites.

**Table 1. Epitopes selected.** Amino acid sequence from epitopes selected of protein EsxG and EsxH.

| Epitope number | Epitope ID | Epitope sequence | Selected epitope | Short name | Selected epitope sequence |
|---|---|---|---|---|---|
| 1 | ep1 | [11]LVASQSAFA[19] | ep1 | G1 | [11]LVASQSAFA[19] |
| 2 | ep2 | [51]FQAAHARFV[59] | ep2—ep3 | G2 | [51]FQAAHARFVAAAAKVN[66] |
| 3 | ep3 | [58]FVAAAAKVN[66] | | | |
| 4 | ep4 | [5]MYNYPAMLG[13] | ep4 | H1 | [5]MYNYPAMLG[13] |
| 5 | ep5 | [65]LVRAYHAMS[73] | ep5—ep6—ep7 | H2 | [65]LVRAYHAMSSTHE[77] |
| 6 | ep6 | [66]VRAYHAMSS[74] | | | |
| 7 | ep7 | [69]YHAMSSTHE[77] | | | |

two RMSD stages previously described. Observing the trajectory, the compactness was mainly due to the N-terminal of the EsxG monomer, which adopted a conformation folded toward helix g1a. The folding of the C and N-terminal from EsxH towards helix h2 (Fig 3A–3C) yielded a more packed structure. RMSF analysis was performed to determine fluctuating regions of the protein, being residues 1 to 7 from N-terminal of EsxG monomer the most fluctuating region (1.465 nm), followed by residue 87 (0.718 nm), residue 88 (0.729 nm), residues 95 to 97 from C-terminal of EsxG (0.684 nm), residues 1 to 6 from N-terminal of EsxH (0.544 nm), and residues 85 to 89 from C-terminal of EsxH (0.523 nm) (S1C Fig). The results showed that EsxG monomer showed greater RMSF values than EsxH.

The secondary structure analysis revealed the following structural changes on the alpha-helix of the G monomer: i) residues 3 to 7, were initially formed a coil and bend structures, then they were structured into alpha-helix, from 378 ns to 374 ns. ii). Residue 18 lost its alpha-helix structure after 10 ns of simulation and was restructured again together with residue 17 from 206 to 382 ns and from 460 ns to 500 ns. iii) Residue 39 lost its alpha-helix structure from 41 to 78 ns and from 379 to 397 ns. iv) The alpha-helix structure elongated from residues 39 to 42 at 2 to 41 ns. v). The g2 helix lengthens at residues 46 to 49 from 78 ns and was maintained up to 500 ns. vi). Residues 74 to 76 changed from turn to alpha-helix structure during 18 to 61 ns and from 126 to 159 ns. vii) Residues 81 to 84 get an intermittent structure between alpha-helix and turn from 122 ns until the end of the simulation. The alpha-helices of monomer EsxH had a more stable secondary structure than the two described before. In contrast, the following changes were observed in the secondary structure of the EsxH monomer: i) residues 5 to 9 of the EsxH chain started the simulation unstructured. They were structured to an alpha-helix from 58 ns and was maintained up to 500 ns of simulation. ii) There was a shortening of helix h1, being residues 21 to 23 the ones that lost their secondary structure after 103 ns of simulation. iii) Residues 38 to 42 were structured in alpha-helix from 24 ns to 138 ns, then the structure fluctuated between alpha-helix and turn until the end of the simulation. iv) Residue 77 to 81 lost its helix structure between 26 ns to 123 ns, later it recovered its structure. v) Residues 82 to 88 lengthened the helix intermittently during the simulation. Despite these changes, the dimer maintained its secondary structure with minimal fluctuations in both monomers (S1D Fig).

Clusterization showed that 80% of the total visited structures along the simulation were sorted in the first eight clusters, although the first cluster was the most populated with 47.9% (S1E Fig). Hydrogen bonds are important to preserve the native secondary and tertiary structure of proteins; 140 intra-protein hydrogen bonds on average were kept through the entire simulation, having a maximum of 169 hydrogen bonds at 320 ns (S1F Fig). A principal component analysis was performed by using Cα atoms to obtain the principal component covariance matrices. Analyzing the first one, it revealed correlated displacements between 1) C and N-terminal from EsxG, 2) N-terminal from EsxG monomer and residues 40 to 55 from EsxG, 3) both terminals from EsxH monomer and N-terminal from EsxG, and 4) the displacements of

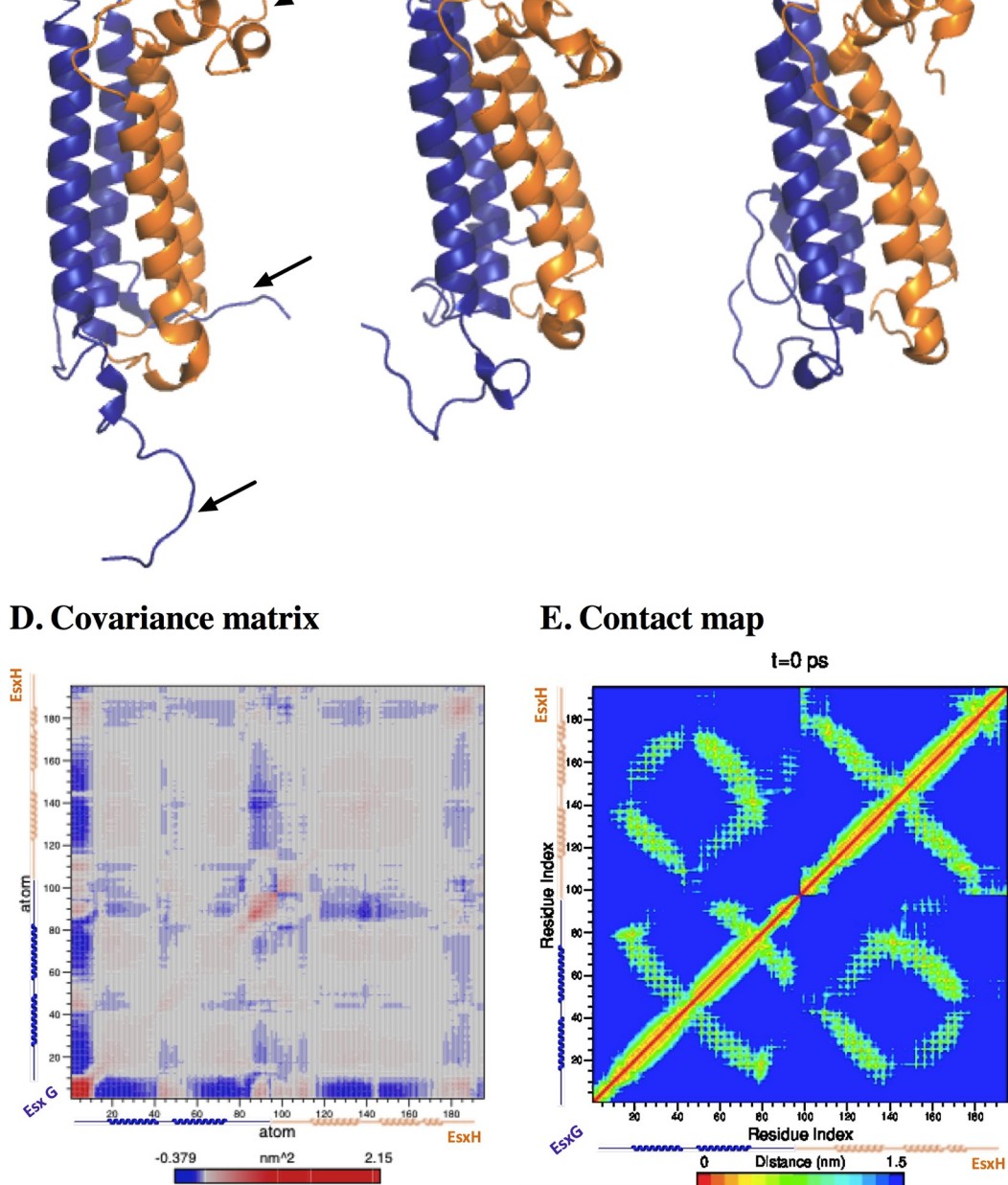

**Fig 3. Global structure behavior from EsxG EsxH dimer at 310 K.** (A). 0 ns of simulation. Arrows indicate regions that underwent folding. (B). 149 ns of simulation. (C). 500 ns of simulation. Blue depicts EsxG monomer; orange depicts EsxH monomer. (D). First principal component covariance matrix. The matrix is color-coded, from red (correlated displacements) to blue (non-correlated displacements). The diagonal line stands for the correlation between the residues paired with themselves; the blue stands for the correlation between each residue pair during the 500 ns simulation. (E). Contact map. The matrix is color-coded, blue (farthest) to red (closest). The diagonal line represents the zero distance between the residues paired with themselves, while spots represent the distances (nm) for each residue pair during the 500 ns simulation.

the four helices of the heterodimer were correlated themselves (Fig 3D). A map contact was performed to evaluate the distance among residues, the folding of the C and N-ends from EsxG and EsxH monomers to their respective helices was shown (Fig 3E). The g2 and h2 helices were the most stable motifs in heterodimer.

## Monomers molecular dynamic simulation

To study the monomer's molecular behavior in solution, MD simulations of each monomer at 310 K were carried out. The EsxG monomer RMSD value stabilized at 57 ns. After this time, one conformational structure with an average of 1.292 nm was presented (S2A Fig) and an average RoG of 1.459 nm (S2B Fig). An average of 62 intra-protein hydrogen bonds, with a maximum of 79 bonds at 185 ns, and a minimum of 47 bonds at 267 ns were found (S2C Fig). N and C-terminal were the most fluctuating regions, with an RMSF value of 0.823 nm in residue 1 and 1.223 nm in residue 97, respectively (S2D Fig). Both terminals folded toward g2 helix (Fig 4A–4C). Secondary structure analysis revealed that residues 15 to 19 fluctuated between a turn structure and alpha-helix after 37 ns, while residues 41 to 43 after 101 ns form a turn structure, residues 63 to 65 after 49 ns fold into a turn structure. Residues 75 and 76 fluctuated between turn and alpha-helix structure after 18 ns, residues 91 to 95 changed from coil to alpha-helix structure between 58 and 98 ns of simulation (S2E Fig). In this case, 75% of the structures were found in the first five clusters, being the first cluster the most populated one with 42.8% (S2F Fig). The first principal component covariance matrix showed correlation movements between the N and C-terminal. While an anticorrelation between the N and C-ends with the g1 and g2 helices was found (Fig 4D).

On the other hand, the EsxH monomer structure stabilized at 66 ns, with an RMSD average of 0.926 nm and a RoG of 1.795 nm (S3A ang S3B Figs, respectively). An average of 63 intra-protein hydrogen bonds was maintained, with a maximum of 85 bonds at 7 ns; and a minimum of 44 bonds at 160 ns and 269 ns (S3C Fig). Residues 19, 29 to 36, 50 to 70, and the C-terminal presented the greatest fluctuation with a maximum RMSF value of 0.848 nm. The least fluctuating regions were residues 9 to 15 with an RMSF between 0.225 and 0.333 nm, and residues 76 to 88 with 0.121 to 0.203 nm (S3D Fig). Helix h1 had an important loss of secondary structure. Moreover, residues 24 to 27 were maintained throughout the simulation whereas residues 28 to 31 were only maintained until 66 ns, but the alpha-helix structure was later lost and restructured at 218 ns. After 188 ns of simulation, residues 34 to 44 were structured in an alpha-helix, which was maintained at the end of the simulation (S3E Fig). Helix h2 maintained a regular stability of secondary structure (Fig 4E–4G). Particularly, 59% of the structures were found in the first 15 clusters, the two initial clusters presented a larger population of structures (S3F Fig). Anticorrelated motions were observed between residues 29 to 41 and 17 to 27, also between residues 27 to 47 and from 47 to 71 (Fig 4H). These results demonstrated that EsxH did not have a stable secondary structure as a monomer, as it was found for EsxG.

## Thermal unfolding

MD simulations supply data related to the unfolding pathway and structural stability at elevated temperatures. Therefore, MD simulations were carried out at different temperatures. Fig 5A shows the initial structure of the dimer. The folding of the N-terminal of EsxG occurred at 5 ns of simulation at 350 K. Residues 20 to 24 from helix h1a evolved to an irregular secondary structure, which remained until 50 ns. At 50 ns of simulation, both ends folded into the EsxG helices. These observations corresponded to the evolution of the RMSD value, which had variations between 0.7 nm and 1.1 nm, from 7 to 30 ns of the simulation. After this time, the structure stabilized and remained close to the mean (0.858 nm) (S4A Fig). The RoG value was close

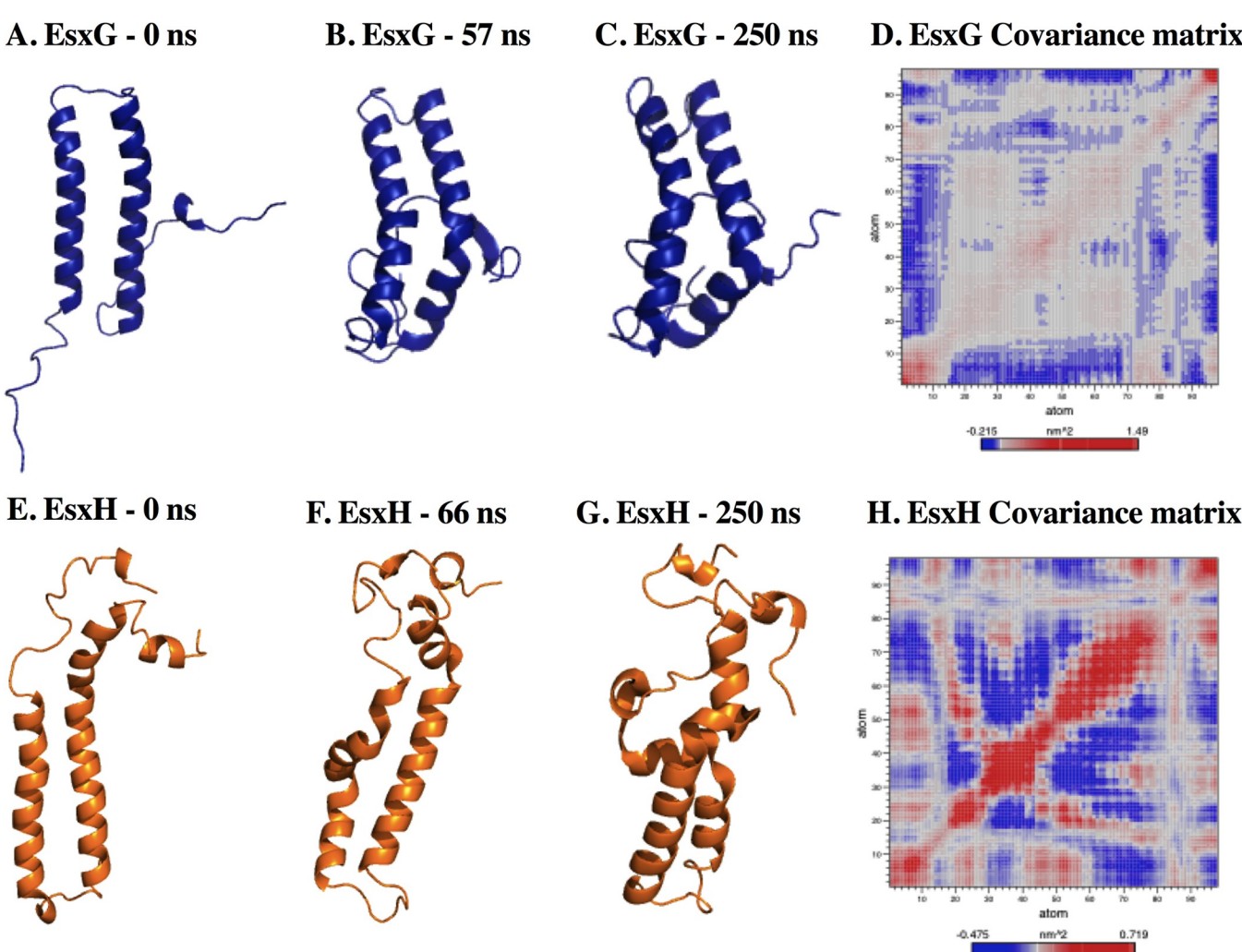

**Fig 4. Global structure behavior of EsxG at 310 K.** (A). at 0 ns of simulation. (B). at 57 ns of simulation. (C). at 250 ns of simulation. (D). First principal component covariance matrix. The matrix is color-coded, from red (correlated displacements) to blue (non-correlated displacements). The diagonal line stands for the correlation between the residues paired with themselves, while the color stands for the correlation between each residue pair during the 250 ns simulation. **Global structure of the first cluster from EsxH at 310 K.** (E). 0 ns of simulation. (F). 66 ns of simulation. (G). 250 ns of simulation. (H). First principal component covariance matrix.

to the mean (1.897 nm) after 10 ns (S4B Fig). The most fluctuating region was the N-terminal of the EsxG monomer with an RMSF value of 0.97 nm, followed by the C-terminal of EsxG and ExH monomer with an RMSF value of 0.84 nm (S4C Fig). The average of the intra-protein hydrogen bonds was 132 (S4D Fig).

The dimer underwent relevant structural changes when applying kinetic energy by increasing temperature. As these changes occurred quickly, just the first 5 ns of the simulations from 400 K to 550 K were considered for analysis.

The average structure at 400 K (Fig 5D), 450 K (Fig 5E), 500 K (Fig 5F), and 550 K (Fig 5G) were calculated. Fig 5G shows a significant loss of native conformation. Regarding RMSD, the average values were 0.887 nm (S5A Fig), 0.810 nm (S5B Fig), 1.292 nm (S5C Fig), and 1.417 nm (S5D Fig), respectively. At 400 K, the radius of gyration was larger (2.142 nm) than 310 K and 350 K (S5E Fig) because the N-terminal of the EsxG monomer unfolded, as did residues 76 and 77 (Thr and His) of the EsxH monomer. Despite this, the secondary structure of the

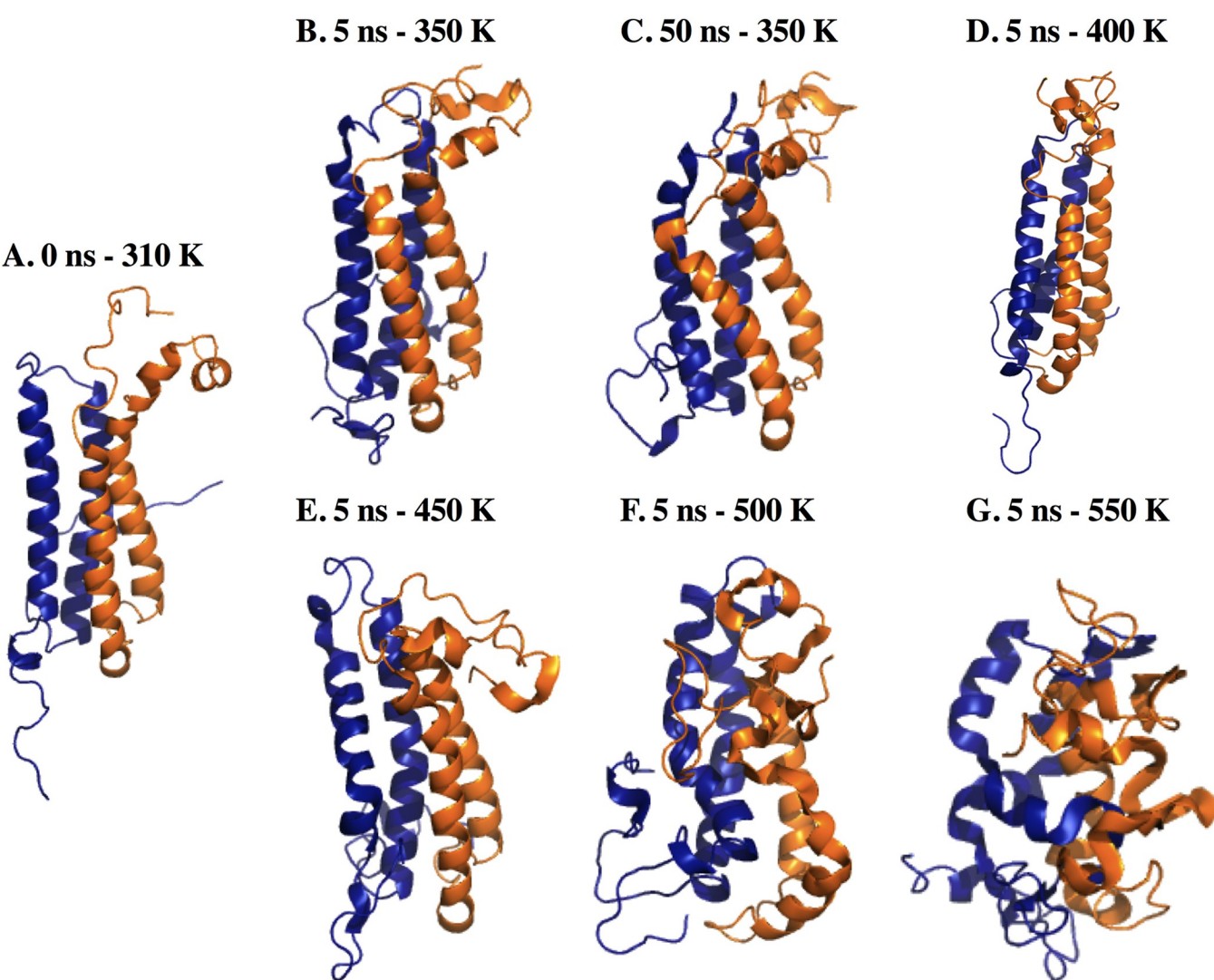

**Fig 5. Average structures from EsxG EsxH dimer thermal unfolding simulations.** (A). 0 ns of simulation. (B). 5 ns of simulation at 350 K. (C). 50 ns of simulation at 350 K. (D). 5 ns of simulation at 400 K. (E). 5 ns of simulation at 450 K. (F). 5 ns of simulation at 500 K. (G). 5 ns of simulation at 550 K.

helix was maintained during the simulation (S6A Fig). However, the intra-protein Hb decreased to 132 (S6E Fig).

At 450 K, the compactness of the dimer was directly proportional to the increase in temperature being RoG average values as follows: 1.962 nm at 450 K (S5F Fig), 1.920 nm at 500 K (S5G Fig), and 1.768 nm at 550 K (S5H Fig). At 450 K, residues 30 to 40 from helix g1 lost their regular secondary structure at 2 ns of simulation. Helix h1 also lost its structure after 2 ns (S6B Fig). Furthermore, helix g1 preserved its structure until 2 ns, but it completely lost its secondary structure. In the opposite way, residues 53 to 60 corresponding to the EsxH monomer kept its structure. Helix g2 has higher stability in terms of secondary structure (S6C Fig).

At 550 K, residues 20 to 30 of the EsxG monomer were maintained during 1.5 ns, and residues 50 to 62 along 5 ns of simulation. EsxH monomer lost its regular secondary structure, turns and a beta secondary structure predominated (S6D Fig).

As for Hydrogen bonds, when the temperature was increased to 450 K the intra-protein Hb average number decreased to 125 (S6F Fig), at 500 K the Hb decreased to 113 (S6G Fig), and at

550 K in the beginning of the simulation decreased to 110 then increased to 145 (S6H Fig). The increase can be explained by interactions generated with the change of motifs in the secondary structure.

## MD simulations: Epitopes and HLA-epitopes complexes

To study the behavior of the epitopes in solution, MD simulations of the selected peptides under physiological conditions were carried out. Furthermore, these epitopes were docked to the MHC-II HLA molecule (PDB ID: 6BIZ [42]) to predict how epitopes are bonded in the HLA, to observe the epitope behavior while forming the complex, and to describe the intermolecular interactions formed in the HLA/epitope complexes. Moreover, MD simulations were carried out from the HLA/epitope complexes to compare them with the epitope solution dynamics.

For epitope G1, one global conformation was observed with a RMSD value of 0.310 nm (S7A Fig). The RMSD of the G2 epitope fluctuated throughout the simulation; the epitope got three conformations, the first one from 0 to 192 ns, the second one from 194 ns to 231 ns, and the third one from 232 ns to the end of the simulation (S7B Fig). H1 epitope adopted a global conformation throughout the simulation, which was maintained with a RMSD of 0.095 nm (S7C Fig). For H2 epitope, a global conformation was found with a RMSD of 0.297 nm (S7D Fig). G2 epitope presented several conformations throughout the simulation. When we compared the RMSD of G1, G2, and H2 epitopes loaded in the HLA, similar behavior for the G1 complex was found, which indicated no change in the conformation (S7A Fig). In G2, H1, and H2 complexes, we observed an increase in the RMSD. G2 complex had one conformation until 234 ns. H1 complex showed three conformations from 0 ns to 32 ns, from 33 ns to 151 ns, and from 152 ns to 250 ns (S7C Fig). In the H2 complex, two conformations were observed from 0 ns to 40 ns, the second one from 50 ns to 250 ns (S7D Fig).

The RoG of the epitopes that remained stable in solution throughout the simulation, showed a mean of 0.833 nm (S7E Fig), 0.806 nm (S7F Fig), 0.709 nm (S7G Fig), and 1.065 nm (S7H Fig) for epitopes G1, G2, H1, and H2, respectively. When the epitopes were coupled to the HLA molecule, the RoG in the G1 remained without major changes (S7E Fig). The Rog in the G2 kept stable up to 237 ns, after this time, the structure lost compactness (S7F Fig). The H1-HLA complex compacted from 18 ns to 132 ns, then lost its compactness and compacts again at 192 ns (S7G Fig). The RoG decreased during the simulation in the H2 indicating compaction of the structure (S7H Fig).

The four epitopes in the solution did not present a regular secondary structure (S8A–S8D Fig). The N and C-terminals of the four epitopes in solution were the regions with the highest RMSF (S9A–S9D Fig). In the arrangement of the epitopes to the HLA, epitopes G1, H1, and H2 continued without having a regular secondary structure. However, the coil and bend motifs remained constant throughout the simulation (Fig 6A, 6C and 6D) (S8E, S8G and S8H Fig). Interestingly, G2 epitope, when loaded into the HLA, took a regular secondary structure, residues 7 to 11 formed an alpha-helix structure that was maintained during the entire simulation (Fig 6B) (S8F Fig). When the epitopes formed a complex with HLA, we observed a decrease in the RMSF value of the N-terminal from G1 (S9A Fig), and in the C-terminal from G2 (S9B Fig) and H1 (S9C Fig), compared to the solvated epitope. In the H2-HLA complex, the RMSF of the N and C-ends were the most fluctuating regions, the central residues of the epitope decreased their RMSF (S9D Fig).

Concerning clusterization of epitopes in solution, we observed that H1 epitope had one predominant conformation visited throughout the simulation (S10C Fig). G1, G2, and H2 epitopes visited at least three conformations during the simulation (S10A, S10B and S10D Fig).

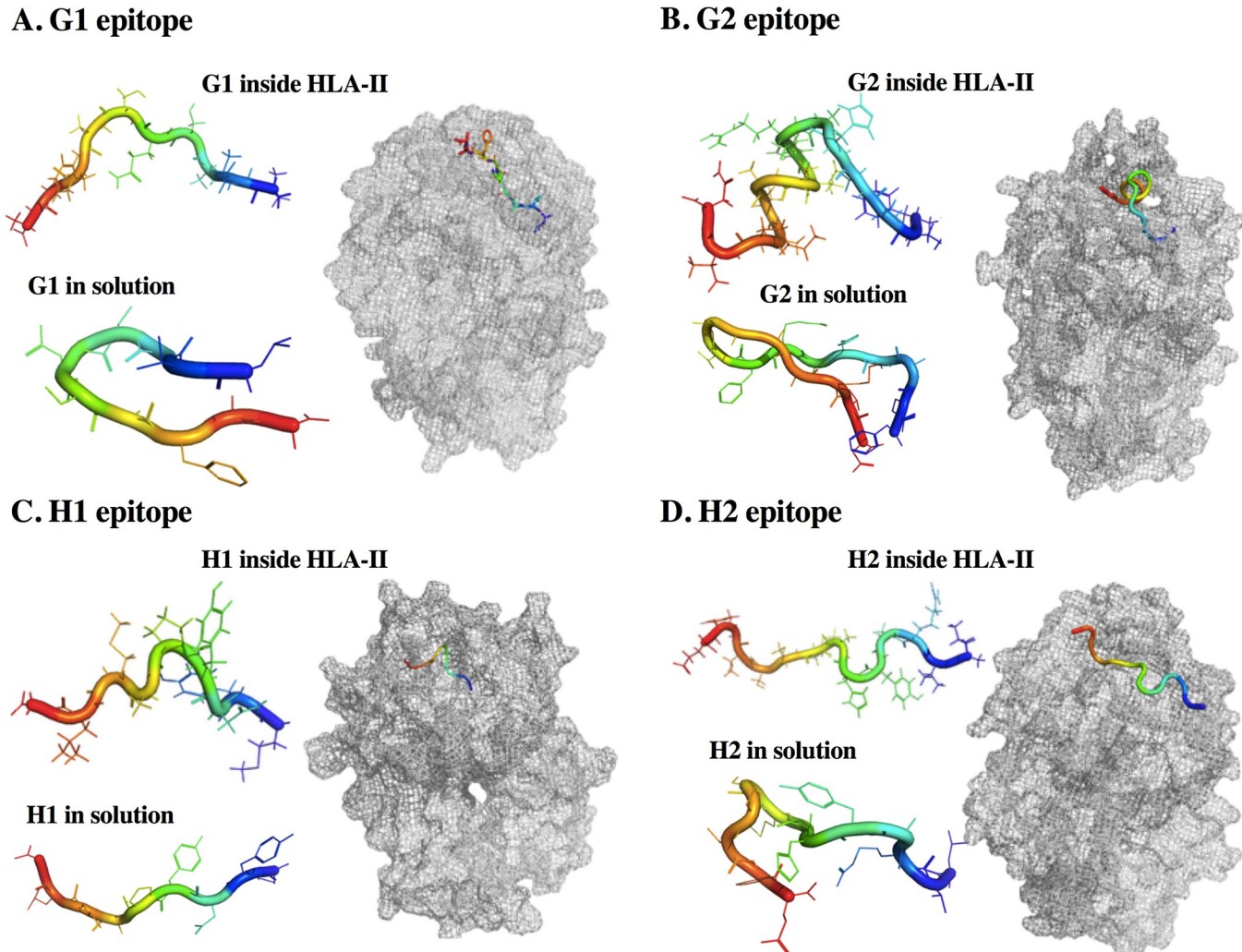

**Fig 6. Average structures from epitopes in solution and HLA-epitopes complexes.** (A). G1 epitope. (B). G2 epitope. (C). H1 epitope. (D). H2 epitope.

When G1 formed the complex with the HLA, we observed one predominant structure (S10E Fig), while G2, H1, and H2 even though it has one predominant structure, visited at least other two conformations during the simulation (S10F–S10H Fig).

We evaluated the total intra-protein bound for each epitope in solution, being 1.68, 2.48, 0.46, and 6.09 for G1, G2, H1, and H2 respectively (S11A–S11D Fig). As we expected, hydrogen bonds increased to 6, 7.75, 4.41, and 9.68 as mean value, respectively, when the epitopes were coupled to the HLA molecule (S11E–S11H Fig). Additionally, we identified the hydrogen bonds with an occupancy greater than 20% to establish which interactions were maintained for the longest time during the simulation (Table 2).

Finally, the solvent-accessible surface area (SASA) of the epitope in solution or forming the complex with HLA molecule was evaluated to determine the residues and the difference in the exposure. The SASA of the epitopes in solution was subtracted from the SASA of the epitopes forming the complex. These values were denominated as ΔSASA (Fig 7). All residues had a ΔSASA greater than zero, so a greater area of exposure to solvent was found in the complex.

**Table 2. Hydrogen bonds occupancy in HLA-epitopes complexes.**

| Epitope | Occupancy | Residue from Epitope –Residue from HLA | Epitope | Occupancy | Residue from Epitope –Residue from HLA |
|---|---|---|---|---|---|
| G1 | 36.44% | V2:m–N82:s (chain B) | H1 | 45.50% | Y2:m–N82:s (chain B) |
| | 22.59% | V2:m–N82:s (chain B) | | 36.85% | G9:s–W61:s (chain B) |
| | 38.18% | A3:m–Q9:s (chain A) | | 22.21% | G9:m–W61:s (chain B) |
| | 40.57% | S4:m–Q9:s (chain A) | | 25.10% | G9:s–Q64:s (chain B) |
| | 33.80% | S4:m–Q9:s (chain A) | | 28.29% | G9:s–N69:s (chain A) |
| | 22.99% | S4:s–Q9:s (chain A) | | 57.93% | G9:s–R71:s (chain B) |
| | 62.77% | F8:m–Q64:s (chain B) | H2 | 85.09% | R3:m–H81:s (chain B) |
| | 32.78% | F8:m–Q64:s (chain B) | | 25.82% | Y5:s–H81:m (chain B) |
| G2 | 27.02% | F1:m–N82:s (chain B) | | 27.89% | H6:m–N62:s (chain B) |
| | 24.98% | F1:m–N82:s (chain B) | | 40.67% | M8:m–Y30:s (chain B) |
| | 28.78% | Q2:m–Q9:s (chain A) | | 56.06% | S9:m–Y47:s (chain B) |
| | 26.06% | H5:s–E55:m (chain A) | | 42.93% | S9:s–R71:s (chain B) |
| | 111.12% | R7:s–N66:s (chain B) | | 67.22% | S10:m–W61:s (chain B) |
| | 49.02% | K14:m–R71:s (chain B) | | 38.34% | H12:s–Y60:m (chain B) |
| | 29.20% | N16:s–R71:s (chain B) | | 225.11% | E13:s–R76:s (chain A) |

m–main chain

s–side chain

## Discussion

In this study, we describe a method to rationally select EsxG EsxH epitopes with the potential to be vaccine candidates for pulmonary TB. Through bioinformatics tools, we selected three epitopes with the best immunogenic characteristics: 1) G1 ([11]LVASQSAFA[19]), 2) H1 ([5]MYNYPAMLG[13]), and 3) H2 ([65]LVRAYHAMSSTHE[77]). Table 3 summarizes the characteristics of the selected epitopes. G1 epitope prediction was consistent with previous work that evaluated the response of CD4[+] T cells from mice infected with Mtb to [6]AHIPQLVASQSA-FAAKAGLM[25] synthetic peptide in *in vitro* assays, associating this peptide to protection [52]. H1 epitope prediction was supported with previous studies, which has shown that the N-terminal region, included epitope [1]MSQIMYNYPAMLGHAGDM[18], was the most immunogenic of the protein [19, 53], but weakly recognized by T cell [18, 19]. Interestingly, [3]QMYNYPAM[11] or [4]IMYNYPAM[11] induced strong specific responses of CD8[+] T cells [54, 55] and were highly promiscuous to bind MHC class I alleles [56]. [3]QMYNYPAM[11] residues are the minimum required to generate cytotoxic response against Mtb in the acute and chronic stage of infection. This cytotoxic effect is through degranulation mechanism and CD95L-mediated apoptosis [57]. In addition, it has been shown that peptide [5]MYNYPAMLG[1] binds with low affinity to A*0201, A*2402, and A*3002, MHC class I alleles [56, 58]. H2 epitope prediction matches with evidence showing that [61]AMEDLVRAYHAMSSTHEA[78] residues induced a strong CD4[+] T cell response [59]. [63]EDLVRAYHAMSSTHEANTMAMMARDTAEAAKWGG[96] residues have been more immunogenic in mice vaccinated with BCG [18]. In contrast, epitope [65]LVRAYHAMS[73] was specific to MHC class I molecules [56]. Our H2 epitope includes H70, H76, and E77 residues, involved in the Zn$^{2+}$ binding [25]. Additionally, we used G2 epitope to validate our selection strategy. It should be noted that G2 contains residues previously associated with protection against TB [52]. Given that MHC molecules have high specificity, the difference in a single amino acid can be crucial for T-cell recognition or ablation [19]. Previous studies have shown that the substitution in one amino acid in EsxH [4]IMYNYPAM[11] epitope, significantly influenced the response and binding to T cells, such as protection against TB in murine models [53, 60].

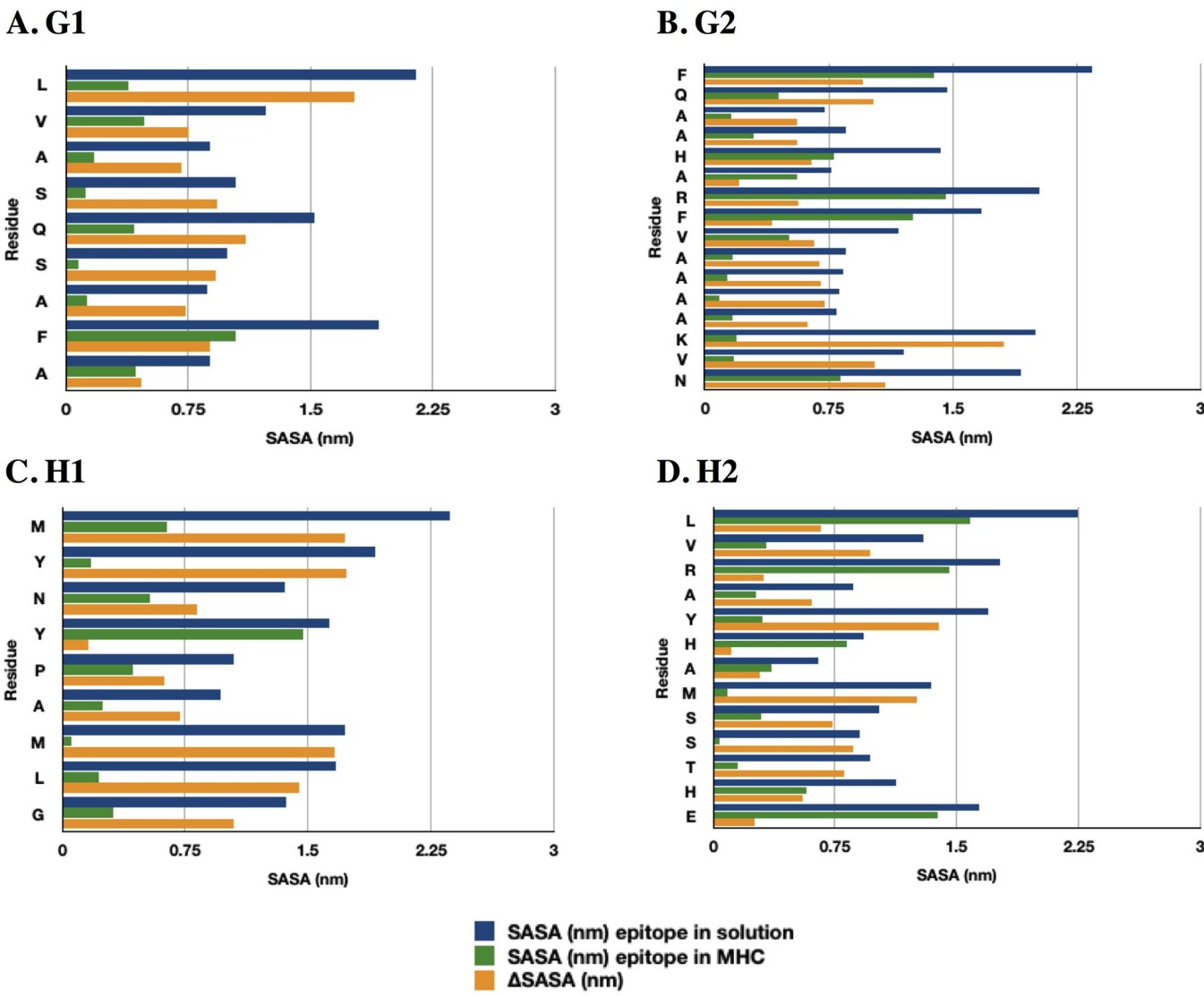

**Fig 7. Solvent accessible surface area (SASA).** (A). G1 epitope. (B). G2 epitope. (C). H1 epitope. (D). H2 epitope. The blue bars depict the SASA from the epitope in solution. The green bars depict the SASA from the epitope in the MHC. The yellow bars depict the ΔSASA, subtracting the SASA of the epitopes in solution from the SASA of the epitopes forming the complex.

**Table 3. Characteristics of the selected epitopes.** Comparison of characteristics from epitopes in solution and when they are presented into HLA complex.

| Analysis | G1 | G2 | H1 | H2 | G1-HLA | G2-HLA | H1-HLA | H2-HLA |
|---|---|---|---|---|---|---|---|---|
| RMSD (nm) | 0.310 | 0.350 | 0.095 | 0.297 | 0.357 | 0.434 | 0.447 | 0.525 |
| RoG (nm) | 0.833 | 0.806 | 0.709 | 1.065 | 2.402 | 2.434 | 2.417 | 2.445 |
| Clusters (to cover 60%) | 13 | 161 | 2 | 3 | 1 | 2 | 4 | 7 |
| Hbonds | 1.68 | 2.48 | 0.46 | 6.09 | 6 | 7.75 | 4.41 | 9.68 |
| B-lineal epitope | Yes | No | Yes | Yes | N/A | N/A | N/A | N/A |
| B-conformational epitope | No | No | Yes | No | N/A | N/A | N/A | N/A |

Molecular Dynamics simulations, a tool of structural vaccinology [4] predicts the stability of EsxG EsxH complex and describes the epitopes. Our results from EsxG EsxH complex at 310 K showed that the N and C-ends of EsxG monomer were the most fluctuating and flexible regions of the complex. However, after 149 ns of simulation, these helices folded into the core of the EsxG monomer (Fig 3A), and this folding remained stable until the end of the simulation (Fig 3B and 3C). Despite the secondary structure analysis revealing certain modifications during the simulation, these changes were not relevant for changing the secondary structure of the dimer. Thus, the fluctuation and flexibility, coupled with their high exposition to the solution and the absence of important changes in the secondary structure of the dimer, could suggest a biological relevance, which is consistent with the conservation of some residues among mycobacterial species, previously demonstrated by Ilghari and collaborators. Analysis of correlation matrix and contact map (Fig 3D and 3E) were in line with the characteristic helix-turn-helix hairpin structure, arranged in an antiparallel way, forming a bundle of 4 helices [25], and we proved that these interactions were held on a 500 ns simulation, even though $Zn^{2+}$ ion was not present in the dimer structure. Although we observed some structural changes during the simulation, these changes did not alter the regular secondary structure in the dimer, which was corroborated by the fact that the hydrogen bonds and the RMSD remained constant, indicating that the complex was highly stable for at least 500 ns. Our results are consistent with previous studies showing that both monomers form a stable 1:1 heterodimeric complex [61]. We observed that the RMSD increased proportionally with the increase in temperature. As expected, the RoG and hydrogen bonds decreased, indicating that the protein begins to undergo denaturation. However, a large part of the secondary structure from the helices was maintained at 450 K (Fig 5A–5G). These results showed that dimer significantly resisted denaturation. It is necessary to form $Zn^{2+}$ ion-protein complex to compare stability and aqueous solution protein behavior described in this work.

The simulation of the EsxG monomer behavior in solution confirms the high fluctuation and flexibility of the N and C-ends (Fig 4A–4C). The PCA results showed that both terminals moved correlatively between them and moved in anticorrelation with the g1 and g2 helices. Additionally, the monomer EsxH showed loss of secondary structure in helix h1 (Fig 4E–4G), indicating that EsxH as a monomer does not have a stable structure. Despite this loss of structure, EsxH maintained its helical structure; however, in the analysis carried out by Lightbody and collaborators, the EsxH monomer did not resist chemical denaturation, so that a molten globule structure was formed. The analysis of the molten globule structure revealed the formation of a hydrophobic mini nucleus. The nucleus was formed by tryptophan residues. In this study, the tryptophan SASA was evaluated to determine the changes in their exposure to the solvent in the EsxH monomer while forming the complex with EsxG protein to evidence the formation of the hydrophobic nucleus and of the molten globule. When we evaluated the tryptophan environment, we found that the solvent exposure of the four tryptophans decreases considerably when the EsxH protein forms a 1:1 complex with the EsxG (Table 4). The results agree with what was previously reported by Lightbody and collaborators, where tryptophan residues W43, W54, and W58 formed a hydrophobic core, located at the interface of the

**Table 4. Solvent accessible surface area in tryptophan residues.**

| Residue | SASA (nm) EsxG EsxH | SASA (nm) EsxH |
|---|---|---|
| W43 | 0.353 | 0.914 |
| W54 | 0.006 | 0.260 |
| W58 | 0.023 | 0.490 |
| W94 | 0.823 | 1.804 |

EsxG-EsxH complex, while tryptophan W94 was more exposed. This hydrophobic nucleus is necessary to induce the partial folding of EsxH and may explain why EsxH as a monomer forms a molten globule. Our data are in accordance with previous results and support the idea that they functionally act as a dimer and not as individual proteins. Likewise, this suggests that they could act as a heterodimer in acidic environments, such as the phagosome, without altering its structure [61].

Unfortunately, we were unable to generate an EsxG EsxH/$Zn^{2+}$ complex for the simulation, since residues involved in metal coordination, as well as the geometry of the metallic site, are not yet well established. This limitation could have influenced the mobility of this region. Ilghari and collaborators [25] reported that the implied $Zn^{2+}$ binding site was located at one end of the complex, near the site where a cleft forms, suggesting a potential site for interaction with a target protein. In our study, we observed the formation of this cleft, at the very beginning of the simulation, however, this structure lost this conformation a few ns after the simulation started (Fig 8), which could indicate that the cleft conformation is restricted to $Zn^{2+}$ presence or other targets. Even with the lack of $Zn^{2+}$, the dimer was still stable during all simulation time.

Because epitopes in solution are highly flexible and disordered, compared to when they are part of a protein complex, conformational variations in these can lead to vaccine failure or success, the reason why we consider it important to predict the stability of epitopes. Our results suggest that the H1 epitope was the most stable, presenting one conformation, suggesting that this was the preferred conformation when it is alone in solution. Despite G1 and H2 epitopes presenting one predominant conformation, they visited at least three conformations. G2 epitope in solution showed great fluctuation having several conformations during the 600 ns simulation, which suggests that it is the one with the most variable conformations in solution. These epitopes bind to multiple alleles of MHC class II molecules, which are useful features for a vaccine candidate in a human population with heterogeneous HLA [62]. HLA class II molecules present antigens to CD4+ T cells, thus being essential for eliminating Mtb [63]. For this reason, we performed a docking to predict the presentation of selected epitopes. DRB1 allele was selected because this binds the majority of known Mtb epitopes; furthermore, it is the allele with the greater surface expression [43]. Our results indicate that when G1 formed a complex with HLA presented a predominant structure.

Despite G2 and H2 as complex had a majority conformation, they presented at least three conformations. Interestingly, G2 epitope folds into an alpha-helix at residues 7 to 12. Remarkably, H1 epitope visited at least four conformations when the HLA complex was formed, which suggests that the binding to the HLA is not completely stable (Fig 6A–6D), which was corroborated by the residues involved in the formation of hydrogen bonds (Table 2).

The three epitopes proposed and considered in the prediction are B cell receptor epitopes too (Table 3). Although our predictions included linear and discontinuous epitopes, most of the B cell epitopes are discontinuous and depend on the conformation. Therefore *in silico* prediction is challenging [3]. H1 epitope contains seven residues that match the discontinuous B cell prediction. Thus, B lymphocytes can produce antibodies against the EsxG and EsxH proteins that are involved in metabolic processes in Mtb, which could be both neutralizing or bactericidal antibodies. Finally, the lack of homology of our epitopes with human and rodent proteins suggests that they might avoid an autoimmune response in future clinical trials. The EsxG and EsxH sequence identities between species implied that the predicted epitopes could be used specifically against several Mtb complex strains. These results support the finding that high conservation between species of Mtb complex and nearly related [19, 64] could be used against nontuberculous strains such as *M. kansasii*, *M. szulgai*, *M. simiae* and *M. gastri*. In agreement with previous studies, mortality related to nontuberculous mycobacteria surpasses

## A. Cleft 0 ns - 310 K    B. Cleft 149 ns - 310 K    C. Cleft 500 ns - 310 K

**Fig 8. Cleft formation in EsxG EsxH dimer at 310 K.** (A). 0 ns. (B). 149 ns. (C). 500 ns. Blue represents the EsxG monomer. Orange represents the EsxH monomer. Residues involved in cleft formation are presented in light blue (EsxG) and yellow (EsxH).

TB infections in developed countries. This added to the difficulty of eradicating and its substantial re-occurrence, finding new therapies is crucial [65, 66].

## Conclusion

This work presented an approach to the rational design of vaccines against pulmonary TB. Using bioinformatics tools and Molecular Dynamics simulations, we selected three epitopes that are B-cell or T-cell epitopes. The epitopes were selected from the EsxG EsxH complex, proteins involved in metal uptake, drug resistance, and immune response evasion makes it an ideal target for rational vaccine development. Thus, the three epitopes proposed in the present work have high potential to be subunit vaccine candidates or conjugated vaccines. These epitopes can be used in conjunction with adjuvants, or as a boost in BCG vaccination to have a robust and highly specific immune response. As they are B-cell epitopes, the generated antibodies could block the binding of metals or proteins involved in the phagolysosome formation, and thus in the prevention of the immune evasion, which could compromise Mtb survival. Although Molecular Dynamics simulations are a very reliable tool to predict molecular behavior, it is mandatory to perform *in vivo* assays with these predicted epitopes to demonstrate their immunogenic activity.

## Supporting information

**S1 Table. MHC class-II binding epitope prediction.** 49 HLA-DR alleles were considered to locate promiscuous binding regions. The output consists of a list of nonameric epitopes capable of binding into a given HLA-DR allele.
(DOCX)

**S1 Fig. A. Root Mean Square Deviation (RMSD) plot from EsxG EsxH complex at 310 K.** Red line represents the RMSD value during 500 ns of simulation. Blue line represents the RMSD average. Shaded bar represents standard deviation. **B. Radius of Gyration (RoG) plot from EsxG EsxH complex at 310 K.** Red line stand for RoG value during 500 ns of simulation.

Blue line stands for the RoG average. Shaded bar represents standard deviation. **C. Root Mean Square Fluctuation (RMSF) plot from EsxG EsxH complex at 310 K.** X axis corresponds to residue number. The blue color depicts the EsxG monomer. The orange color depicts the EsxH monomer. **D. Secondary structure plot from EsxG EsxH complex at 310 K.** Residues 1 to 97 represents EsxG monomer, residues 98 to 194 represents EsxH monomer. **E. Clusterization plot from EsxG EsxH complex at 310 K.** X axis stand for the cluster ID. Y axis stand for the number of structures in the cluster. **F. Hydrogen bonds plot from EsxG EsxH complex at 310 K.** Black line stand for total hydrogen bonds during 500 ns of simulation.
(TIF)

**S2 Fig. A. Root Mean Square Deviation (RMSD) plot from EsxG monomer at 310 K.** Red line depict the RMSD value during 250 ns of simulation. Blue line depicts the RMSD average. Shaded bar represents standard deviation. **B. Radius of Gyration (RoG) plot from EsxG monomer at 310 K.** Red line represents RoG value during 250 ns of simulation. Blue line represents the RoG average. Shaded bar represents standard deviation. **C. Hydrogen bonds plot from EsxG monomer at 310 K.** Black line stand for total hydrogen bonds during 250 ns of simulation. **D. Root Mean Square Fluctuation (RMSF) plot from EsxG monomer at 310 K.** X axis corresponds to residue number. **E. Secondary structure plot from EsxG monomer at 310 K.** X axis corresponds to residue number. **F. Clusterization plot from EsxG monomer at 310 K.** X axis represents the cluster ID. Y axis represents the number of structures.
(TIF)

**S3 Fig. A. Root Mean Square Deviation (RMSD) plot from EsxH monomer at 310 K.** Red line depict the RMSD value during 250 ns of simulation. Blue line depicts the RMSD average. Shaded bar represents standard deviation. **B. Radius of Gyration (RoG) plot from EsxH monomer at 310 K.** Red line represents RoG value during 250 ns of simulation. Blue line represents the RoG average. Shaded bar represents standard deviation. **C. Hydrogen bonds plot from EsxH monomer at 310 K.** Black line represents total hydrogen bonds during 250 ns of simulation. **D. Root Mean Square Fluctuation (RMSF) plot from EsxH monomer at 310 K.** X axis corresponds to residue number. **E. Secondary structure plot from EsxH monomer at 310 K.** X axis corresponds to residue number. **F. Clusterization plot from EsxH monomer at 310 K.** X axis represents the cluster ID. Y axis represents the number of structures.
(TIF)

**S4 Fig. A. Root Mean Square Deviation (RMSD) plot from EsxG EsxH complex at 350 K.** Red line stand for the RMSD value during 50 ns of simulation. Blue line stands for the RMSD average. Shaded bar represents standard deviation. **B. Radius of Gyration (RoG) plot from EsxG EsxH complex at 350 K.** Red line depict RoG value during 50 ns of simulation. Blue line depicts the RoG average. Shaded bar represents standard deviation. **C. Root Mean Square Fluctuation (RMSF) plot from EsxG EsxH complex at 350 K.** X axis corresponds to residue number. The blue color depicts the EsxG monomer. The orange color represents the EsxH monomer. **D. Hydrogen bonds plot from EsxG EsxH complex at 350 K.** Black line represents total hydrogen bonds during 50 ns of simulation.
(TIF)

**S5 Fig. A. Root Mean Square Deviation (RMSD) plot from EsxG EsxH complex at 400 K.** Red line represents the RMSD value during 5 ns of simulation. Blue line represents the RMSD average. Shaded bar represents standard deviation. **B. Root Mean Square Deviation (RMSD) plot from EsxG EsxH complex at 450 K.** Red line stand for the RMSD value during 5 ns of simulation. Blue line stands for the RMSD average. Shaded bar stands for standard deviation. **C. Root Mean Square Deviation (RMSD) plot from EsxG EsxH complex at 500 K.** Red line

depict the RMSD value during 5 ns of simulation. Blue line depicts the RMSD average. Shaded bar depicts standard deviation. **D. Root Mean Square Deviation (RMSD) plot from EsxG EsxH complex at 550 K.** Red line represents the RMSD value during 5 ns of simulation. Blue line represents the RMSD average. Shaded bar represents standard deviation. **E. Radius of Gyration (RoG) plot from EsxG EsxH complex at 400 K.** Red line stand for RoG value during 5 ns of simulation. Blue line stands for the RoG average. Shaded bar stands for standard deviation. **F. Radius of Gyration (RoG) plot from EsxG EsxH complex at 450 K.** Red line represents RoG value during 5 ns of simulation. Blue line represents the RoG average. Shaded bar represents standard deviation. **G. Radius of Gyration (RoG) plot from EsxG EsxH complex at 500 K.** Red line depict RoG value during 5 ns of simulation. Blue line depicts the RoG average. Shaded bar represents standard deviation. **H. Radius of Gyration (RoG) plot from EsxG EsxH complex at 550 K.** Red line represents RoG value during 5 ns of simulation. Blue line represents the RoG average. Shaded bar represents standard deviation.
(TIF)

**S6 Fig. A. Secondary structure plot from EsxG EsxH complex at 400 K.** Residues 1 to 97 depict EsxG monomer, residues 98 to 194 depict EsxH monomer. **B. Secondary structure plot from EsxG EsxH complex at 450 K.** Residues 1 to 97 depict EsxG monomer, residues 98 to 194 represents EsxH monomer. **C. Secondary structure plot from EsxG EsxH complex at 500 K.** Residues 1 to 97 stand for EsxG monomer, residues 98 to 194 stand for EsxH monomer. **D. Secondary structure plot from EsxG EsxH complex at 550 K.** Residues 1 to 97 represents EsxG monomer, residues 98 to 194 represents EsxH monomer. **E. Hydrogen bonds plot from EsxG EsxH complex at 400 K.** Black line depict total hydrogen bonds during 5 ns of simulation. **F. Hydrogen bonds plot from EsxG EsxH complex at 450 K.** Black line depict total hydrogen bonds during 5 ns of simulation. **G. Hydrogen bonds plot from EsxG EsxH complex at 500 K.** Black line stand for total hydrogen bonds during 5 ns of simulation. **H. Hydrogen bonds plot from EsxG EsxH complex at 550 K.** Black line stand for total hydrogen bonds during 5 ns of simulation.
(TIF)

**S7 Fig. A. Root Mean Square Deviation (RMSD) plot from G1 HLA complex at 310 K.** Red line represents the RMSD value from the G1-HLA complex during 250 ns of simulation. Black line represents the RMSD value from the G1 epitope in solution during 250 ns of simulation. Blue line represents the RMSD average. Shaded bar represents standard deviation. **B. Root Mean Square Deviation (RMSD) plot from G2 HLA complex at 310 K.** Red line depict the RMSD value from the G2-HLA complex during 250 ns of simulation. Black line depicts the RMSD value from the G2 epitope in solution during 250 ns of simulation. Blue line depicts the RMSD average. Shaded bar represents standard deviation. **C. Root Mean Square Deviation (RMSD) plot from H1 HLA complex at 310 K.** Red line stand for the RMSD value from the H1-HLA complex during 250 ns of simulation. Black line stands for the RMSD value from the H1 epitope in solution during 250 ns of simulation. Blue line stands for the RMSD average. Shaded bar stands for standard deviation. **D. Root Mean Square Deviation (RMSD) plot from H2 HLA complex at 310 K.** Red line represents the RMSD value from the H2-HLA complex during 250 ns of simulation. Black line represents the RMSD value from the H2 epitope in solution during 250 ns of simulation. Blue line represents the RMSD average. Shaded bar represents standard deviation. **E. Root Mean Square Deviation (RMSD) plot from G1 HLA complex at 310 K.** Red line depict the RMSD value from the G1-HLA complex during 250 ns of simulation. Green line depicts the RMSD value from the G1 epitope in solution during 250 ns of simulation. Blue line depicts the RMSD average. Shaded bar represents standard deviation. **F. Root Mean Square Deviation (RMSD) plot from G1 HLA complex at 310 K.**

Red line stand for the RMSD value from the G2-HLA complex during 250 ns of simulation. Green line stands for the RMSD value from the G2 epitope in solution during 250 ns of simulation. Blue line stands for the RMSD average. Shaded bar represents standard deviation. **G. Root Mean Square Deviation (RMSD) plot from H1 HLA complex at 310 K.** Red line depict the RMSD value from the H1-HLA complex during 250 ns of simulation. Green line depicts the RMSD value from the H1 epitope in solution during 250 ns of simulation. Blue line depicts the RMSD average. Shaded bar represents standard deviation. **H. Root Mean Square Deviation (RMSD) plot from H2 HLA complex at 310 K.** Red line represents the RMSD value from the H2-HLA complex during 250 ns of simulation. Green line represents the RMSD value from the H2 epitope in solution during 250 ns of simulation. Blue line represents the RMSD average. Shaded bar represents standard deviation.
(TIF)

**S8 Fig. A. Secondary structure plot from G1 epitope in solution at 310 K.** Residues 1 to 9 stand for the G1 epitope LVASQSAFA. **B. Secondary structure plot from G2 epitope in solution at 310 K.** Residues 1 to 16 stand for the G2 epitope FQAAHARFVAAAAKVN. **C. Secondary structure plot from H1 epitope in solution at 310 K.** Residues 1 to 9 represents the H1 epitope MYNYPAMLG. **D. Secondary structure plot from H2 epitope in solution at 310 K.** Residues 1 to 13 represents the H2 epitope LVRAYHAMSSTHE. **E. Secondary structure plot from G1 epitope in HLA at 310 K.** Residues 1 to 9 depict the G1 epitope LVASQSAFA. **F. Secondary structure plot from G2 epitope in HLA at 310 K.** Residues 1 to 16 depict the G2 epitope FQAAHARFVAAAAKVN. **G. Secondary structure plot from H1 epitope in HLA at 310 K.** Residues 1 to 9 depict the H1 epitope MYNYPAMLG. **H. Secondary structure plot from H2 epitope in HLA at 310 K.** Residues 1 to 13 depict the H2 epitope LVRAY-HAMSSTHE.
(TIF)

**S9 Fig. A. Root Mean Square Fluctuation (RMSF) plot from G1 epitope in solution, and in HLA complex at 310 K.** X axis corresponds to residue number, residues 1 to 9 stand for the G1 epitope LVASQSAFA. Blue color stand for G1 epitope in solution. Black color stand for G1 epitope in complex with HLA. **B. Root Mean Square Fluctuation (RMSF) plot from G2 epitope in solution, and in HLA complex at 310 K.** X axis corresponds to residue number, residues 1 to 16 depict the G2 epitope FQAAHARFVAAAAKVN. Blue color represents G1 epitope in solution. Black color depicts G1 epitope in complex with HLA. **C. Root Mean Square Fluctuation (RMSF) plot from H1 epitope in solution, and in HLA complex at 310 K.** X axis corresponds to residue number, residues 1 to 9 depict the H1 epitope MYNY-PAMLG. Blue color depicts G1 epitope in solution. Black color depicts G1 epitope in complex with HLA. **D. Root Mean Square Fluctuation (RMSF) plot from H2 epitope in solution, and in HLA complex at 310 K.** X axis corresponds to residue number, residues 1 to 13 represents the H2 epitope LVRAYHAMSSTHE. Blue color represents G1 epitope in solution. Black color represents G1 epitope in complex with HLA.
(TIF)

**S10 Fig. A. Clusterization plot from G1 epitope in solution at 310 K.** X axis depict the cluster ID. Y axis depict the number of structures. **B. Clusterization plot from G2 epitope in solution at 310 K.** X axis represents the cluster ID. Y axis represents the number of structures. **C. Clusterization plot from H1 epitope in solution at 310 K.** X axis stand for the cluster ID. Y axis stand for the number of structures. **D. Clusterization plot from H2 epitope in solution at 310 K.** X axis represents the cluster ID. Y axis represents the number of structures. **E. Clusterization plot from G1 epitope in HLA at 310 K.** X axis depict the cluster ID. Y axis depict

the number of structures. **F. Clusterization plot from G2 epitope in HLA at 310 K.** X axis stand for the cluster ID. Y axis stand for the number of structures. **G. Clusterization plot from H1 HLA in solution at 310 K.** X axis represents the cluster ID. Y axis represents the number of structures. **H. Clusterization plot from H2 epitope in HLA at 310 K.** X axis stand for the cluster ID. Y axis stand for the number of structures.
(TIF)

**S11 Fig. A. Hydrogen bonds plot from G1 in solution at 310 K.** Orange line depict total hydrogen bonds during 250 ns of simulation. **B. Hydrogen bonds plot from G2 in solution at 310 K.** Orange line depict total hydrogen bonds during 250 ns of simulation. **C. Hydrogen bonds plot from H1 in solution at 310 K.** Orange line represents total hydrogen bonds during 250 ns of simulation. **D. Hydrogen bonds plot from H2 in solution at 310 K.** Orange line represents total hydrogen bonds during 250 ns of simulation. **E. Hydrogen bonds plot from G1 in HLA at 310 K.** Black line stand for total hydrogen bonds during 250 ns of simulation. **F. Hydrogen bonds plot from G2 in HLA at 310 K.** Black line stand for total hydrogen bonds during 250 ns of simulation. **G. Hydrogen bonds plot from H1 in HLA at 310 K.** Black line represents total hydrogen bonds during 250 ns of simulation. **H. Hydrogen bonds plot from H2 in HLA at 310 K.** Black line represents total hydrogen bonds during 250 ns of simulation.
(TIF)

## Author Contributions

**Conceptualization:** Constanza Estefania Martinez-Olivares, Rogelio Hernández-Pando, Edgar Mixcoha.

**Data curation:** Constanza Estefania Martinez-Olivares, Edgar Mixcoha.

**Formal analysis:** Constanza Estefania Martinez-Olivares, Edgar Mixcoha.

**Funding acquisition:** Edgar Mixcoha.

**Investigation:** Constanza Estefania Martinez-Olivares, Edgar Mixcoha.

**Methodology:** Constanza Estefania Martinez-Olivares, Edgar Mixcoha.

**Resources:** Edgar Mixcoha.

**Visualization:** Constanza Estefania Martinez-Olivares, Edgar Mixcoha.

**Writing – original draft:** Constanza Estefania Martinez-Olivares, Edgar Mixcoha.

**Writing – review & editing:** Constanza Estefania Martinez-Olivares, Rogelio Hernández-Pando, Edgar Mixcoha.

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
