## [Decision Letter · Decision Letter 0]

25 Oct 2022

PONE-D-22-18255

In silico EsxG·EsxH Rational Epitope Selection: Candidate Epitopes for Vaccine Design against Pulmonary Tuberculosis

PLOS ONE

Dear Dr. Mixcoha,

Thank you for submitting your manuscript to PLOS ONE. After careful consideration, we feel that it has merit but does not fully meet PLOS ONE’s publication criteria as it currently stands. Therefore, we invite you to submit a revised version of the manuscript that addresses the points raised during the review process.

We look forward to receiving your revised manuscript.

Kind regards,

Angelo A. Izzo

Academic Editor

PLOS ONE

“This paper was supported by grant 413957, CVU 406611 from the National Council for Science and Technology of Mexico (Consejo Nacional de Ciencia y Tecnología-CONACyT). Project number 21-2019 and 13-2020 by LANCAD supercomputing center.”

“This paper was supported by grant 413957, CVU 406611 from the National Council for Science and Technology of Mexico (Consejo Nacional de Ciencia y Tecnología-CONACyT). Project number 21-2019 and 13-2020 by LANCAD supercomputing center.

“This paper was supported by grant 413957, CVU 406611 from the National Council for Science and Technology of Mexico (Consejo Nacional de Ciencia y Tecnología-CONACyT). Project number 21-2019 and 13-2020 by LANCAD supercomputing center.”

Reviewers' comments:

Reviewer's Responses to Questions

**Comments to the Author**

1. Is the manuscript technically sound, and do the data support the conclusions?

Reviewer #1: Yes

Reviewer #2: Yes

2. Has the statistical analysis been performed appropriately and rigorously? 

Reviewer #1: Yes

Reviewer #2: N/A

3. Have the authors made all data underlying the findings in their manuscript fully available?

Reviewer #1: Yes

Reviewer #2: Yes

4. Is the manuscript presented in an intelligible fashion and written in standard English?

Reviewer #1: Yes

Reviewer #2: Yes

5. Review Comments to the Author

Reviewer #1: The following manuscript describes the rational design of novel vaccine epitopes targeting pulmonary tuberculosis through a suite of predictive servers and molecular dynamics simulations. As a proof of concept, the authors used the EsxG:EsxH complex as template, an heterodimer involved into the bacteria viability and homeostasis (i.e. zinc uptake). The two proteins are also involved into antibiotic resistance mechanisms.

The group first selected 21 epitopes from the proteins EsxG and EsxH that could bind to MHC class II using ProPred and NetMHCII 2.3 platforms. They predicted additional 13 linear B cell epitopes from the same protein complex using BepiPred 2.0 (8 continuous epitopes) and DiscoTope 2.0 (5 discontinuous epitopes) - Figure 1. The epitopes were subjected to peptide-protein docking experiments using CABS-Dock and FlexPepDock to predict their binding conformations into MHC class II molecules, in particular HLA-DRB1. Only seven out of all epitopes were selected for further studies (Table 1), namely G1, G2, H1 and H2. On the second large section of the manuscript, the authors evaluated the predicted folding and structural stability of (1) the heterodimeric complex and the two distinct monomers EsxG and EsxH [Figures 2-4, 7 and S1-S6] and (2) the epitopes and HLA-DRB1-epitope complexes [Figures 5-6, S7-S11, and Table 2] using multiple molecular dynamics simulations. The four epitopes did not fold into specific secondary structures in solution or in complex. They identified the 4 epitopes forming complexes with HLA-DRB1. In their discussion, the authors supported their bioinformatic observations for epitopes G1, H1, H2 (Table 3) with external experimental validation throughout the scientific literature.

The manuscript is well-documented and information-rich. Some sentences throughout the manuscript are hard to grasp and should be re-written for clarity (e.g. lines 49-51). I encourage the authors to review and edit the document.

Please consider the following minor corrections;

(1) Mycobacterium tuberculosis should be italicized.

(2) Typo – line 453 RMFS

Reviewer #2: The paper by Martinez-Olivares et al. describes a rational design of epitope-based peptide vaccines by using bioinformatics and structural vaccinology tools, focusing on the testing of EsxG·EsxH complex. The ESX systems from Mycobacterium tuberculosis are very well-known factors for the secretion of highly immunogenic proteins that seem of key importance for bacterial survival and growth. I appreciated the fact that, even if the authors use already investigated tools and approaches, they explore the fusion of EsxG and EsxH proteins as candidate target for TB vaccine development, never tested before in this conjugated version, especially for the presence of EsxH protein.

The pipeline is well described and the results appropriately report the potentiality to be considered as subunit vaccine candidates or conjugated vaccines, even if the validation phase is mandatory for the evaluation of the immunogenicity. The conclusions convey the main points of the manuscript and the methodology is exhaustively described in the Methods section.

I would recommend the authors to address my concerns here below before final pubblication in order to provide to the scientific community this further step in the field of TB vaccine development.

- Inside the Introduction part, the authors should integrate the bibliography related to the rational vaccine design through immunobioinformatics tools and computational chemistry approaches. They report only two references (not so recent) while the literature offers more innovative and updated examples of new vaccine development pipeline such as the one reported in Sunita et al., 2019 (doi:10.1080/21645515.2019.1670035) and some specific examples applied for example to COVID-19 vaccine (doi: 10.1093/bib/bbab403).

- In Results section, the authors mention the role of the MTB in provoking intracellular infection, justifying the importance of MHC class II epitope prediction. For the sake of completeness, the authors should also report specific results related also to the extracellular infection that Mtb is able to cause and the consequent main immunological actors dynamics involved in this perspective. I would suggest then to insert new plots or piece of evidence in such view, for example focusing on MHC class I epitope prediction.

- The "Thermal unfolding simulations" is missing the bibliography. Please insert it.

- Even if the Methods are well described, I would suggest the authors to insert inside the manuscript a summary figure reporting all the steps of the rational design approach applied.

6. PLOS authors have the option to publish the peer review history of their article (what does this mean?). If published, this will include your full peer review and any attached files.

Reviewer #1: No

Reviewer #2: **Yes: **Giulia Russo

---

## [Author Response · Author response to Decision Letter 0]

17 Nov 2022

Reviwer #1.

Thank you for all your valuable comments about our work, these are helpful to improve the quality of our work.

1. Reviewer #1 Suggest checking in detail sentences that are difficult to follow.

Author response: We revised our manuscript in detail to find sentences that were difficult to follow, as you suggested. In the reviewed manuscript, we found some sentences that are already changed and highlighted in yellow in the file named “Revised Manuscript with Track Changes”

2. Reviewer #1 recommend check all typos throughout the manuscript. 

Author response: After reviewing the text, all the typos found are already corrected. 

Reviewer #2

We want to thank all your comments, we found them helpful to make our clearer and more objective in the submitted manuscript. 

1. Reviewer #2 recommended incorporating recent articles related to rational vaccine design in Introduction Section.

Author response: We have incorporated a new bibliography in the Introduction Section. In the file named “Revised Manuscript with Track Changes,” you can corroborate it.

2. Reviewer #2 advises incorporating reports related to extracellular infection and evidence of epitopes MHC-I prediction.

Author response: As required, information about the results related to the extracellular Mtb infection related to the humoral immune response was added in the Results section with three references that support this information

3. Reviewer #2 pointed out a missed bibliography in Thermal unfolding simulations.

Author response: We have checked that missed cite and incorporated it in the indicated section.

4. Reviewer #2 suggests inserting a summary figure inside the manuscript reporting all the steps of the rational design approach applied.

Author response: We have designed a summary figure as the reviewer suggested. (Fig 1 in the file “Revised Manuscript with Track Changes”)

---

## [Decision Letter · Decision Letter 1]

2 Jan 2023

PONE-D-22-18255R1In silico EsxG·EsxH Rational Epitope Selection: Candidate Epitopes for Vaccine Design against Pulmonary TuberculosisPLOS ONE

Dear Dr. Mixcoha,

Thank you for submitting your manuscript to PLOS ONE. After careful consideration, we feel that it has merit but does not fully meet PLOS ONE’s publication criteria as it currently stands. Therefore, we invite you to submit a revised version of the manuscript that addresses the points raised during the review process.

We look forward to receiving your revised manuscript.

Kind regards,

Angelo A. Izzo

Academic Editor

PLOS ONE

Journal Requirements:

Reviewers' comments:

Reviewer's Responses to Questions

**Comments to the Author**

1. If the authors have adequately addressed your comments raised in a previous round of review and you feel that this manuscript is now acceptable for publication, you may indicate that here to bypass the “Comments to the Author” section, enter your conflict of interest statement in the “Confidential to Editor” section, and submit your "Accept" recommendation.

Reviewer #2: (No Response)

Reviewer #3: All comments have been addressed

2. Is the manuscript technically sound, and do the data support the conclusions?

Reviewer #2: (No Response)

Reviewer #3: Yes

3. Has the statistical analysis been performed appropriately and rigorously? 

Reviewer #2: (No Response)

Reviewer #3: N/A

4. Have the authors made all data underlying the findings in their manuscript fully available?

Reviewer #2: (No Response)

Reviewer #3: Yes

5. Is the manuscript presented in an intelligible fashion and written in standard English?

Reviewer #2: (No Response)

Reviewer #3: Yes

6. Review Comments to the Author

Reviewer #2: Dear Authors,

it's a bit hard for me to catch the new advancements and revisions to indicate if you have adequately addressed my comments. Could you please provide a new version with highlighted or marked in bold parts concerning my comments?

Thank you.

Reviewer #3: Please add the full name of" EsxG·EsxH" protein complex in the beginning of the study and then through out the manuscript the abbreviation can be used

7. PLOS authors have the option to publish the peer review history of their article (what does this mean?). If published, this will include your full peer review and any attached files.

Reviewer #2: **Yes: **Giulia Russo

Reviewer #3: No

---

## [Author Response · Author response to Decision Letter 1]

3 Feb 2023

Reviewer #2: Dear Authors,

it's a bit hard for me to catch the new advancements and revisions to indicate if you have adequately addressed my comments. Could you please provide a new version with highlighted or marked in bold parts concerning my comments?

Thank you.

Author response

Reviewer #2. We regret being unclear in our previous response. In order to make all suggestions easier to understand and read, we are currently underlining and highlighting them in the "Revised Manuscript with Track Changes" file.

On October 25th, we received the very first comments and advice from Reviewers of our submitted work titled “In silico EsxG·EsxH Rational Epitope Selection: Candidate Epitopes for Vaccine Design against Pulmonary Tuberculosis” . In this way, reviewer #2's comments stated:

Reviewer #2: The paper by Martinez-Olivares et al. describes a rational design of epitope-based peptide vaccines by using bioinformatics and structural vaccinology tools, focusing on the testing of EsxG·EsxH complex. The ESX systems from Mycobacterium tuberculosis are very well-known factors for the secretion of highly immunogenic proteins that seem of key importance for bacterial survival and growth. I appreciated the fact that, even if the authors use already investigated tools and approaches, they explore the fusion of EsxG and EsxH proteins as candidate target for TB vaccine development, never tested before in this conjugated version, especially for the presence of EsxH protein.

The pipeline is well described and the results appropriately report the potentiality to be considered as subunit vaccine candidates or conjugated vaccines, even if the validation phase is mandatory for the evaluation of the immunogenicity. The conclusions convey the main points of the manuscript and the methodology is exhaustively described in the Methods section.

I would recommend the authors to address my concerns here below before final pubblication in order to provide to the scientific community this further step in the field of TB vaccine development.

- Inside the Introduction part, the authors should integrate the bibliography related to the rational vaccine design through immunobioinformatics tools and computational chemistry approaches. They report only two references (not so recent) while the literature offers more innovative and updated examples of new vaccine development pipeline such as the one reported in Sunita et al., 2019 (doi:10.1080/21645515.2019.1670035) and some specific examples applied for example to COVID-19 vaccine (doi: 10.1093/bib/bbab403).

Author response

We modified the manuscript in order to update cites about new techniques and immunoinformatics, also we mentioned both cite that Reviewer #2 advised to us. That changes are present in Pag. 2 lines 32 to 38 in the Introduction section of the final Manuscript. It is worth mentioning that references 5, 6, 7, 8, 9, 10 and, 39 were added as recommended by Reviewer #2.

- In Results section, the authors mention the role of the MTB in provoking intracellular infection, justifying the importance of MHC class II epitope prediction. For the sake of completeness, the authors should also report specific results related also to the extracellular infection that Mtb is able to cause and the consequent main immunological actors dynamics involved in this perspective. I would suggest then to insert new plots or piece of evidence in such view, for example focusing on MHC class I epitope prediction.

Author response

We have included the reports that reviewer #2 pointed out. These modifications are present in the final Manuscript, Pag. 10, lines 218 to 224 in the Results section.

- The "Thermal unfolding simulations" is missing the bibliography. Please insert it.

Author response

This reference is placed already in Pag. 7, line 148.

- Even if the Methods are well described, I would suggest the authors to insert inside the manuscript a summary figure reporting all the steps of the rational design approach applied.

Author response

A new figure (Fig 1 in the final manuscript) was made for this proposal. In Pag. 9, lines 195 to 201 contain explanation about that Summary Figure.

-*-*-*-*-*-*-*-*-*-*-*-*-*-*-*-*-*

Reviewer #3: Please add the full name of" EsxG·EsxH" protein complex in the beginning of the study and then through out the manuscript the abbreviation can be used

Author response

Thank you very much for your comments, we addressed the new change that you suggested to us this time.

We have completed the name of the EsxG·EsxH" protein complex in the abstract section (Pag. 1, lines 3 and 4) and, in the Introduction section, Pag. 3, lines 60 and 61 in the final Manuscript.

---

## [Decision Letter · Decision Letter 2]

29 Mar 2023

In silico EsxG·EsxH Rational Epitope Selection: Candidate Epitopes for Vaccine Design against Pulmonary Tuberculosis

PONE-D-22-18255R2

Dear Dr. Mixcoha,

We’re pleased to inform you that your manuscript has been judged scientifically suitable for publication and will be formally accepted for publication once it meets all outstanding technical requirements.

Kind regards,

Angelo A. Izzo

Academic Editor

PLOS ONE

Additional Editor Comments (optional):

Reviewers' comments:

Reviewer's Responses to Questions

**Comments to the Author**

1. If the authors have adequately addressed your comments raised in a previous round of review and you feel that this manuscript is now acceptable for publication, you may indicate that here to bypass the “Comments to the Author” section, enter your conflict of interest statement in the “Confidential to Editor” section, and submit your "Accept" recommendation.

Reviewer #1: All comments have been addressed

Reviewer #2: All comments have been addressed

2. Is the manuscript technically sound, and do the data support the conclusions?

Reviewer #1: Yes

Reviewer #2: Yes

3. Has the statistical analysis been performed appropriately and rigorously? 

Reviewer #1: N/A

Reviewer #2: Yes

4. Have the authors made all data underlying the findings in their manuscript fully available?

Reviewer #1: Yes

Reviewer #2: Yes

5. Is the manuscript presented in an intelligible fashion and written in standard English?

Reviewer #1: Yes

Reviewer #2: Yes

6. Review Comments to the Author

Reviewer #1: The authors have addressed the comments I made in the first round of reviews, and those of other reviews in the subsequent forms.

Reviewer #2: I'm quite satisfied about the author response and edits to my comments provided by the authors. Hence, I would recommend the manuscrupit for final publication.

7. PLOS authors have the option to publish the peer review history of their article (what does this mean?). If published, this will include your full peer review and any attached files.

Reviewer #1: No

Reviewer #2: **Yes: **Giulia Russo

---

## [Editor Report · Acceptance letter]

10 Apr 2023

PONE-D-22-18255R2 

*In silico* EsxG·EsxH Rational Epitope Selection: Candidate Epitopes for Vaccine Design against Pulmonary Tuberculosis 

Dear Dr. Mixcoha:

I'm pleased to inform you that your manuscript has been deemed suitable for publication in PLOS ONE. Congratulations! Your manuscript is now with our production department. 

Kind regards, 

on behalf of

Dr. Angelo A. Izzo 

Academic Editor

PLOS ONE